# A retrieval-specific mechanism of adaptive forgetting in the mammalian brain

Pedro Bekinschtein [1,5], Noelia V. Weisstaub[2,5], Francisco Gallo [2,5], Maria Renner[2] & Michael C. Anderson[3,4]

Forgetting is a ubiquitous phenomenon that is actively promoted in many species. How and whether organisms' behavioral goals drive which memories are actively forgotten is unknown. Here we show that processes essential to controlling goal-directed behavior trigger active forgetting of distracting memories that interfere with behavioral goals. When rats need to retrieve particular memories to guide exploration, it reduces later retention of other memories encoded in that environment. As with humans, this retrieval-induced forgetting is competition-dependent, cue-independent and reliant on prefrontal control: Silencing the medial prefrontal cortex with muscimol abolishes the effect. cFos imaging reveals that prefrontal control demands decline over repeated retrievals as competing memories are forgotten successfully, revealing a key adaptive benefit of forgetting. Occurring in 88% of the rats studied, this finding establishes a robust model of how adaptive forgetting harmonizes memory with behavioral demands, permitting isolation of its circuit, cellular and molecular mechanisms.

[1] Instituto de Biología Celular y Neurociencias, Facultad de Medicina, UBA-CONICET, Paraguay 2155 3er piso, C1121ABG Buenos Aires, Argentina. [2] Grupo de Neurociencia de Sistemas, Departamento de Fisiología, Facultad de Medicina, University of Buenos Aires, Paraguay 2155 7mo piso, C1121ABG Buenos Aires, Argentina. [3] MRC Cognition and Brain Sciences Unit, University of Cambridge, Cambridge CB2 7EF, UK. [4] Behavioural and Clinical Neurosciences Unit, University of Cambridge, Cambridge CB2 3EB, UK. [5] Present address: Institute for Translational and Cognitive Neuroscience, Institute of Cognitive Neuroscience and Translational (INECO), National Council of Science and Technology (CONICET) Universidad Favaloro, Pacheco de Melo 1860, C1126AAB Buenos Aires, Argentina. These authors contributed equally: Pedro Bekinschtein, Noelia V. Weisstaub. Correspondence and requests for materials should be addressed to P.B. (email: pbekinschtein@favaloro.edu.ar) or to M.C.A. (email: michael.anderson@mrc-cbu.cam.ac.uk)

Darwin proposed that "mental powers" such as memory and attention are subject to natural selection and evolved along with species' physical attributes[1]. Although evolution might seem to have favored good memory, many animals, including insects and mammals, possess active forgetting mechanisms[2–11]. For example, *Drosophila* actively forgets olfactory fear conditioning via dopaminergic neurons that undo learning that supports conditioned behavior[5,6]; and rats actively forget object locations via regulated endocytosis of GluA2/AMPAR receptors, gradually weakening hippocampal synapses[4,9]. Active forgetting mechanisms such as these may have evolved because memory loss enables other adaptive traits[2,4,10]. For example, active forgetting may enable organisms to adapt their behavior flexibly in a changing environment by judiciously forgetting memories that become irrelevant[2,7,11]. To be adaptive, however, such a process requires an updating signal indicating a memory's irrelevance. Although the origins of such a signal are unknown, one possibility is that it derives from the prefrontal cortex. In mammals, the prefrontal cortex enables flexible behavior[12–15] via control mechanisms that suppress habitual responses that might otherwise dominate goal-directed action[12,15]. If the prefrontal cortex suppresses competing responses, it may also suppress competing memories, initiating a signal that triggers active forgetting, tuning this process to an organism's behavioral demands.

Here we tested the existence of an adaptive forgetting mechanism triggered by control processes that support flexible behavior in mammals. We build on extensive research in humans indicating that the act of remembering can cause forgetting[16–28]. For example, we have found that when people retrieve a past event, other memories that compete with and hinder retrieval are more likely to be forgotten[16]. This retrieval-induced forgetting occurs for a broad range of stimuli and contexts[17–22]. Evidence suggests that retrieval-induced forgetting arises because trying to retrieve a specific memory triggers inhibitory control mechanisms mediated by the lateral prefrontal cortex that focus retrieval on goal-relevant traces by suppressing distracting memories[20,23]. Because memory systems throughout the animal kingdom confront the need to selectively retrieve goal-relevant memories, we hypothesized that the control process that inhibits competing memories may be conserved across species. This inhibition process may function as an irrelevance signal that triggers adaptive forgetting of competing memories, ensuring an efficient memory system. If so, retrieval-induced forgetting may provide an important model of adaptive forgetting.

To test whether control processes trigger adaptive forgetting, we studied exploratory behavior in rodents. Rats innately prefer novel objects to familiar ones and, in displaying this preference, reveal memory for the familiar object[29,30]. For example, when presented with a novel and a familiar object, they readily approach the novel item and investigate it. This preference to explore novel objects is part of an exploratory drive present from infancy[31–33], and arises in diverse species[31,34]. Building on this behavior, we modified procedures from human studies of retrieval-induced forgetting[35,36] to test whether instinctively triggered retrieval entrained by exploratory behavior can drive retrieval-induced forgetting in rodents. We further sought to establish critical cognitive parallels between rodents and humans necessary to infer an inhibitory control process as the origin of forgetting, and to show that prefrontal mechanisms underlie these parallels. Establishing such parallels would point to a species-general prefrontal control mechanism linking behavioral flexibility to adaptive forgetting; linking this forgetting to an instinctive behavior would offer a model of how this mechanism serves adaptive behavior. We found that when rats retrieved past events, it caused substantial and enduring forgetting of competing memories, and that this active forgetting required control processes supported by the prefrontal cortex.

## Results

**Retrieval practice induces forgetting of competing memories.** Rats' preference to explore novel objects requires engaging retrieval to distinguish which objects are familiar, so that exploration can be directed elsewhere[29]. We capitalized on this tethering of innate behavior and cognition to test whether remembering a prior encounter with one object caused rats to forget other objects seen in the same setting. To achieve this, we adapted the spontaneous object recognition procedure[29–31] to include three phases: encoding, retrieval practice, and test. During encoding, rats incidentally associated an environment (e.g., an arena) to two objects (objects A and B). We first exposed them to two copies of object A, followed by two copies of object B (Fig. 1a). These two sessions occurred in the same arena, 20 min apart (see Supplementary Methods and Supplementary Figure 3), associating, for the rat, the arena to memories of exploring the two objects. Exposing A and B in separate sessions minimized inter-competitor associations, which reduce retrieval-induced forgetting in humans[37,38].

Thirty minutes after exploring the objects, we led rats to repeatedly retrieve their memory of one object (randomly selected). During this retrieval practice task, we exposed rats to one of the objects (e.g., object A) three times, each time paired with a second object they had never seen in that arena (objects X, Y, and Z). Importantly, the rats previously had viewed these paired objects (e.g., X) in a different arena, making them familiar, but novel to the current context. Because both A and the paired object were familiar, we considered any preference rats had for exploring the contextually novel object as evidence that they had retrieved a context-specific memory for exploring A. Fifteen minutes separated each retrieval practice exposure. Our aim was to examine whether repeatedly retrieving object A led rats to forget seeing object B. We based this prediction on the assumption that during each retrieval practice, the arena's environmental features would initially activate memories of exploring A and B, triggering retrieval competition. Selectively retrieving a context-specific memory of A, therefore, should require rats to resolve competition from their trace for B, which we expected would engage an inhibitory control mechanism mediated by the prefrontal cortex. We hypothesized that during retrieval practice, this prefrontal control mechanism would modulate representations of B, likely in the dorsal hippocampus[4,39,40] to suppress that trace (Fig. 1b), triggering active forgetting of B, similar to retrieval-induced forgetting in humans[16–28]. We further posited that this forgetting, once triggered by inhibition during retrieval practice, would persist without further involvement of the medial prefrontal cortex (mPFC), reflecting an enduring disruption.

To measure adaptive forgetting, we tested rats' object recognition 30 min after the last retrieval practice trial. We tested recognition of both A (hereinafter, the practiced object) and B (the competitor) across two test trials. On each trial, we paired one of these "old" items with an entirely novel object (e.g., objects C and D) and scored the time rats spent exploring the old object vs. the novel object (Fig. 1a). To quantify how well rats recognized the old item, we computed a discrimination index reflecting the bias in the time they spent exploring the novel item instead of the old one (Fig. 1c). According to our hypothesis, if the prior retrieval practice triggered inhibitory control to disrupt memory for the competitor, the resulting forgetting should be observed on this final test as a tendency for rats to now perceive the competitor as though it were new. Behaviorally, rats should reveal

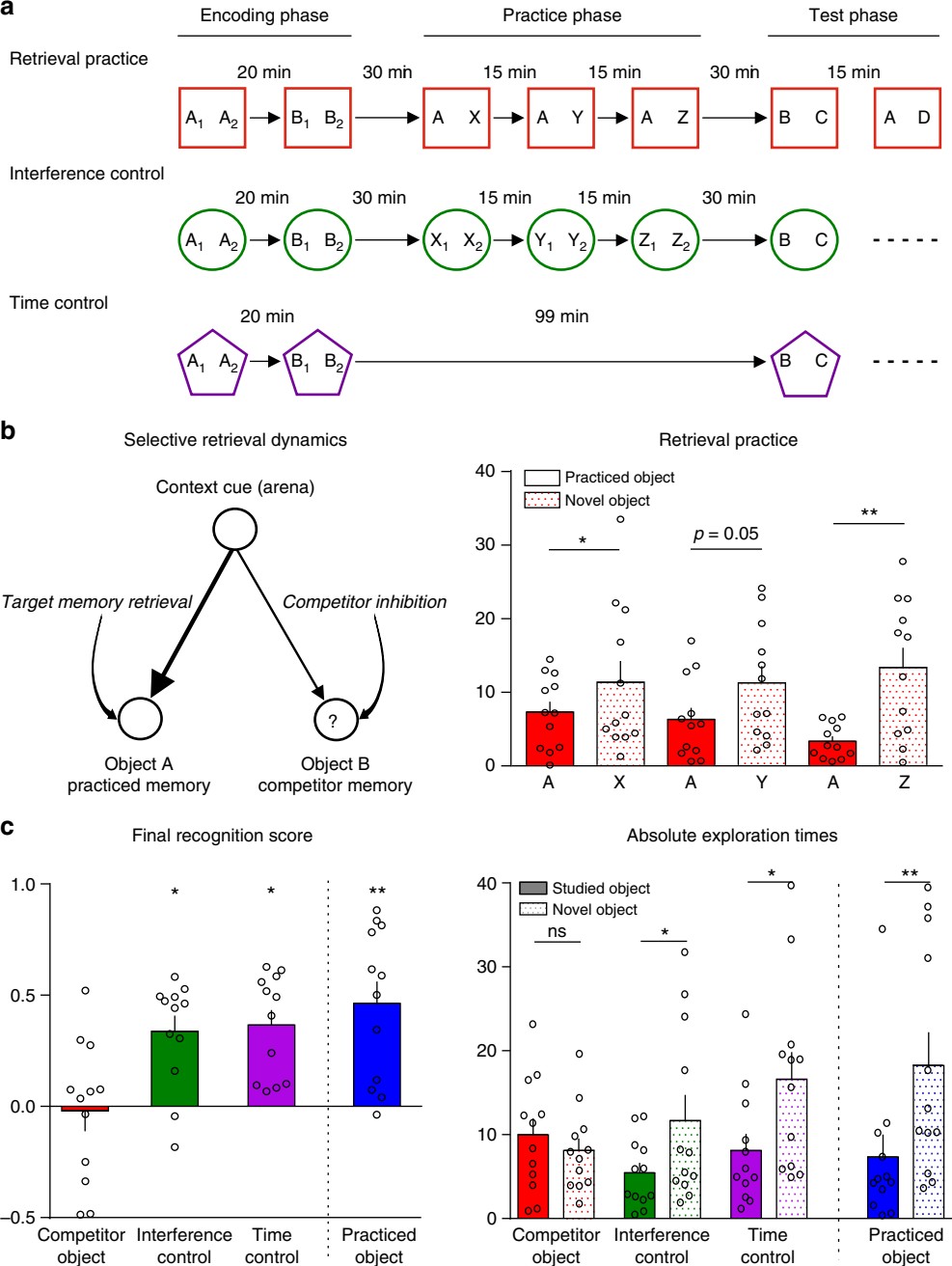

**Fig. 1** Retrieval practice induces forgetting of competing memories. **a** The three phases of Experiment 1 (encoding, practice, test) for the three conditions. During encoding, rats associated arena contexts (different arenas depicted as distinct colored shapes) to two copies of each of two objects (distinct letters indicate different objects; numbers indicate copies). After encoding, rats retrieved one of the objects three times during practice via a novelty preference procedure in which they were exposed to an encoded object with a contextually novel object. The final test paired the competitor (e.g., B) or the practiced item (e.g., A) with a fully novel object. The interference and time control conditions replaced practice with novel object encoding sessions and time in the home cage, respectively. **b** (Left). The assumed retrieval dynamics during practice. Efforts to retrieve object A during exploration elicit competition from the memory for B, triggering control processes to inhibit B. Inhibition's persisting aftereffects hinder retrieval of B during test. (Right) During practice, the number of seconds that animals explored the practiced and contextually novel objects during the three trials. Individual values used to calculate the mean and SEM are presented as dots. Practiced object exploration diminished across trials, indicating increased familiarity. Values are means ± SEM. *$p = 0.033$, **$p = 0.003$, one-tailed paired $t$ test; effect sizes ($d$): $d = 0.83$ (X vs. A); $d = 0.90$ (Y vs. A); $d = 4.34$ (Z vs. A). **c** (Left) Discrimination index means ± SEM calculated from the exploration times during test. *$p = 0.024$ ($t_{33} = 3.08$), $d = 1.12$ (RP vs. IC); *$p = 0.0130$ ($t_{33} = 3.32$), $d = 1.21$ (RP vs. TC); $p = 0.0013$ ($t_{33} = 4.15$), $d = 1.52$ (RP vs. RP+), Bonferroni post hoc comparisons after a repeated-measures ANOVA, $n = 12$ animals. Discrimination index = (novel-object exploration time − familiar object exploration time)/total exploration time. Recognition of the competitor object B was impaired relative to interference and time controls. (Right) Absolute exploration times in seconds during test. The novel and the competitor objects were explored equally after retrieval practice, but not in the other conditions. Values are means ± SEM. *$p < 0.05$ and **$p < 0.01$, NS = not significant. Paired $t$ test, $n = 12$ animals

this perception through a reduced preference for exploring the truly novel object (e.g., C) instead of the competitor; the rat should view both as similarly new. To test whether rats preferred the novel object less, we compared the performance to two control conditions. The first control eliminated retrieval practice by returning rodents to their home cages after encoding A and B, for the duration of the retrieval practice phase, controlling for the effect of time on memory (hereinafter, the time control). In the second control condition, after encoding, we inserted rats into the arena the same number of times as in the retrieval practice condition, but instead exposed them to two copies of contextually novel objects on each trial (e.g., two copies of X on trial 1; then two copies of Y; then two copies of Z). Because this control never re-exposed items on the intervening practice trials, selective retrieval of A or B was unlikely. This novel-object encoding task controlled for repeated exposures to the arena, and to objects, the encoding of which might interfere with memory for A and B (hereinafter, the interference control). All rats participated in these three conditions (Fig. 1a), except in Experiments 6 and 7. We predicted that, on the final test, rats would recognize the competitor objects more poorly in the retrieval practice condition than in the time control or interference control conditions. This pattern would indicate that retrieval disrupts competing memories, and that forgetting exceeds memory loss from passive decay over time, or from encoding novel objects.

Experiment 1 tested this prediction. First, we confirmed that animals explored both copies of the object equally during the encoding sessions (Supplementary Figure 1A). During the subsequent retrieval practice phase, we further confirmed that rats preferred to explore the contextually novel object (Fig. 1b, right panel, practiced object vs. X: *$p = 0.033$, $t_{11} = 2,04$; practiced object vs. Y: $p = 0.05$, $t_{11} = 1.78$; practiced object vs. Z: **$p = 0.0033$, $t_{11} = 3.73$; see also Supplementary Table 1). Rats clearly preferred the contextually novel object, suggesting that during retrieval practice, they remembered the old object, leading them to explore the object never seen in this arena. This pattern grew stronger over the three repetitions. Together, these findings suggest that rats encoded both the practiced object and the competitor, and that they then retrieved memories of the practiced objects throughout the practice phase, as we had hoped.

Our main concern was whether repeatedly retrieving the practiced objects led rats to forget the competitors, which we evaluated on the final test. Strikingly, on this test, rats did not prefer novel items, exploring the competitor and novel objects to the same degree (Fig. 1c, $p = 0.83$, $t_{11} = 0.22$, in a one-sample $t$ test against "0"). Rats thus showed no indication of remembering the competitor. This forgetting was not caused by the passage of time between the encoding and test phases: in our time control condition, the same delay intervened, yet rats strongly preferred exploring the novel object ($p < 0.0001$, $t_{11} = 5.967$ in a one-sample $t$ test against "0"). Confirming this difference, rats remembered old objects better in the time control than in the retrieval practice condition (Fig. 1c, left panel *$p < 0.05$ competitor vs. time control, Bonferroni post hoc comparisons after one-way analysis of variance (ANOVA), **$p_{(ANOVA)} = 0.0013$, $F_{(3, 33)} = 6.63$). In contrast, rats tended to remember the repeatedly practiced objects better, though not significantly.

Rats might have forgotten the competitor for reasons other than retrieval practice. For example, during retrieval practice, we also exposed rats to the arena repeatedly, handled them more, and allowed them to encode three contextually novel objects as distractors. The time control condition lacked these features. To test whether these features caused forgetting, we tested recognition in the interference control condition (Fig. 1a). In this condition, we replaced retrieval practice with three contextually novel-object-encoding sessions, and during these sessions, rats

explored the objects as they had during the encoding phase (Supplementary Figure 1B and Supplementary Table 1). Despite encoding three new object-arena associations during these sessions, rats later recognized test objects as well as in the time control condition (Fig. 1c, left panel). On the test, rats preferred exploring novel objects in the interference control, showing good memory ($p = 0.0005$, $t_{11} = 4,81$, in a one-sample $t$ test, against "0"). Critically, rats recognized competitors in the retrieval practice condition more poorly than in this interference control (Fig. 1c, left panel, *$p < 0.05$, Bonferroni comparisons after one-way ANOVA, **$p_{(ANOVA)} = 0.0013$, $F_{(3, 33)} = 6.63$). Absolute exploration times also showed that rats preferred the novel object in the interference and time control conditions, but not in the retrieval practice condition (Fig. 1c, right panel, *$p < 0.05$, **$p < 0.01$, paired $t$ test, $n = 12$ rats).

Thus, retrieving memories to guide exploration led rats to forget competing traces more than would be expected from decay over time or from encoding interfering objects. These findings support a retrieval-specific adaptive forgetting mechanism.

**Adaptive forgetting reflects competitor inhibition.** The forgetting of competing memories in Experiment 1 may not have been caused by active inhibition. Repeated retrieval practice simply may have strengthened rats' memory for the practiced object, yielding a strong memory associated to the arena context; as a result, when rats later re-entered the arena during the test, memories of the practiced object may have dominated retrieval, making it harder to recall the competitor. Retrieval failure arising during the test from this hypothetical blocking might be mistaken for competitor inhibition. Research with humans indicates that blocking is not sufficient to explain retrieval-induced forgetting[20,21,23,35], but this process has never been examined in rats.

To test this blocking hypothesis, we had rats explore a new object (hereinafter, a new competitor) after retrieval practice ended. According to blocking, rats should forget these new objects for the same reasons they forget competitors encoded before retrieval practice: increased interference on final test. Specifically, at test, entering the arena should remind rats of their most dominant memories from that environment (i.e., the practiced objects), interfering with retrieval of weaker memories. This difficulty should affect all weaker memories regardless of when they were learned. In contrast, if retrieval practice suppresses competitors via inhibitory control, then inhibition should only affect memories encoded before retrieval practice (competitors); they are the only memories that could interfere with retrieval practice and require suppression. This latter prediction has been confirmed in humans, who do not show retrieval-induced forgetting for memories encoded after retrieval practice[41]. To distinguish blocking from active inhibition, our new encoding phase after retrieval practice allowed rats to explore two copies of a novel object (Fig. 2a and Supplementary Figure 4). Rats thus associated a new competitor object J with the arena after practice had ended, in addition to the original competitor object B, encoded before retrieval practice (Fig. 2b). Because of this new encoding step, in the final test, we measured whether rats could recognize the competitor (Fig. 2c) and also the new competitor (Fig. 2c) (test order varied across rats). We also inserted this new encoding phase into both the time control and interference control conditions, providing additional baseline object memories that matched our new competitor objects in how recently they were encoded (Fig. 2a).

During encoding, rats explored the competitor and the new competitor (Supplementary Table 2) enough to encode both (Supplementary Table 2). Confirming this, rats recognized these objects to a similar degree in each of our control conditions

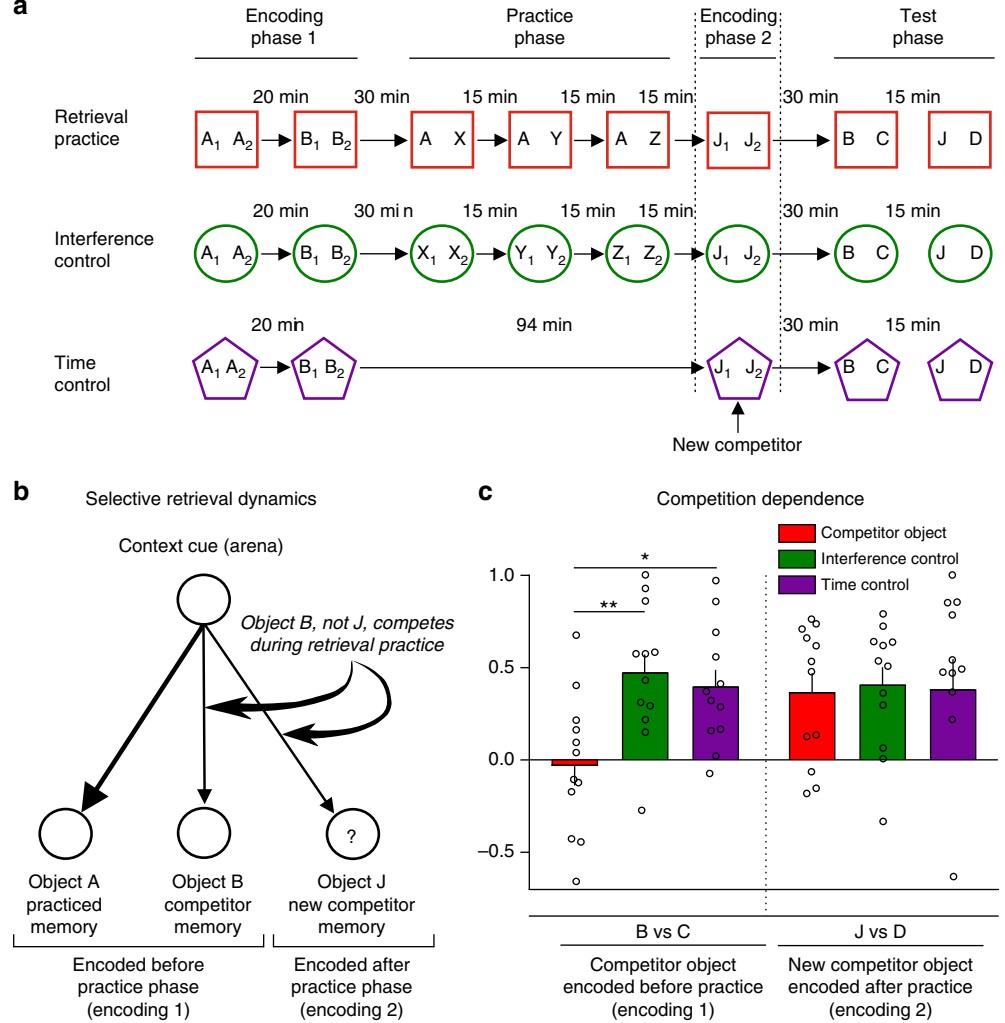

**Fig. 2** Forgetting is competition-dependent. **a** The four phases used in Experiment 2. A new encoding phase (encoding 2) introduced two copies of a new object J after the practice phase. **b** The assumed retrieval dynamics during practice. Efforts to retrieve object A would trigger control processes to inhibit B, but not the new competitor memory J encoded post practice. **c** Discrimination index means ± SEM during test. Individual values used to calculate the mean and SEM are presented as dots. We found a significant interaction for condition × time of encoding, **$p = 0.005$ ($t_{22} = 3.68$), $d = 1.52$ (RP vs. IC); and *$p = 0.019$ ($t_{22} = 3.14$), $d = 1.12$ (RP vs. TC). Bonferroni post hoc comparisons after a repeated-measures ANOVA, $n = 12$ animals

(Fig. 2c and Supplementary Table 3). Critically, however, retrieval-induced forgetting varied for the competitor and the new competitors (Fig. 2c; repeated-measures ANOVA, interaction time of encoding × condition ($F_{(2, 22)} = 3.473$, $p = 0.048$, $n = 12$ rats). Competitors encoded before retrieval practice suffered retrieval-induced forgetting relative to baseline items in the two control conditions (Figs. 2c, *$p < 0.05$, time control vs. competitor object, $t_{22} = 3.14$; interference control vs. competitor object, **$p < 0.05$, $t_{22} = 3.68$; comparisons after Bonferroni correction), but new competitors did not (Fig. 2c). As in Experiment 1, rats showed no indication of recognizing competitors encoded before retrieval practice ($p = 0.78$, $t_{11} = 0.28$, in a one-sample $t$ test, against "0"). These findings suggest that forgetting does not reflect blocking from a dominant memory. If this had been true, blocking should have affected the new competitors as well. Rather, the arena context must already be associated to a competitor prior to retrieval practice for the competitor to require adaptive forgetting via inhibitory control, paralleling findings in humans[41].

We also tested whether rats' performance during retrieval practice predicted later forgetting of competitors. During retrieval practice trials, directing exploration to the contextually novel object requires that rats remember the old item, which should benefit from the ability to resolve competition via inhibitory control. If so, retrieval practice success (i.e., strong novelty preference) should predict later forgetting of competitors. The data confirmed this hypothesis: the stronger rats' preference for the novel object (across all practice trials), the worse their retention of the competitor on the final test, $r = -0.67$, $p = 0.02$ ($-0.94$, $-0.04$) (robust correlation, 95% confidence interval). The same correlation arose when we combined data from Experiments 1–5 ($N = 63$), $r = -0.40$, $p = 0.001$, ($-0.58$, $-0.19$). In contrast, retrieval practice performance during Experiment 2 did not predict final recognition of new competitors, which could not have competed during retrieval practice, $r = 0.08$, ($-0.57$, $0.72$). Thus, rats' ability to direct exploration to novel objects benefits from forgetting competing memories, pointing to a key advantage of forgetting.

**Adaptive forgetting generalizes across contexts**. Retrieval practice in Experiments 1 and 2 may have weakened the competing memory's association to the arena. Weakening this association may have made it harder for rats to recognize the

competitor upon re-entering the arena during the test. Alternatively, retrieval practice may have inhibited the competing memory itself, independent of its association to the arena. This latter alternative should be more likely if the current phenomenon reflects the process observed in humans[18,20,21,23]. Human research on retrieval-induced forgetting supports a property known as cue-independence[23]. Cue-independence refers to the tendency for forgetting to generalize to a variety of cues. This property suggests that retrieval practice does not simply undo the association between the cue and competing memories; rather, it disrupts the competing memories, making them generally less accessible. This pattern constitutes strong evidence for memory inhibition not readily explained by interference.

Experiment 3 tested for cue-independence by examining whether forgetting generalizes across contexts. For these experiments, rats explored the competitor twice. On each occasion, the competitor appeared in a different arena with distinct characteristics. We sought to have the rats associate the competitor to two contexts, with only one shared by the practiced object. Our goal was to see whether the rat performing retrieval practice in one arena context (arena 1) would induce forgetting of the competitor object not only in that context, but also in the second arena (arena 2). To test this possibility, rats encoded objects A and B in arena 1, and also object B in arena 2 (Fig. 3a). We adopted special measures encouraging rats to treat the repetitions of B as two encounters with the same object (as opposed to experiences with distinct copies) (for details see Fig. 3a and Supplementary Tables 4–5). After encoding, rats performed retrieval practice in arena 1, as in Experiments 1 and 2, with one change: between repetitions, we placed rats into the unpracticed arena (arena 2) and exposed them to novel objects. We interleaved sessions in arena 2 to match the amount and the recency of rats' exposure to the arenas. Critically, during the final test, we tested the competitor in the practiced arena (arena 1) for one group, and in the unpracticed arena (arena 2) for another (Fig. 3a). Within each group, every animal also participated in the interference and time control conditions. Encoding and test phases were the same for all conditions.

During encoding and retrieval practice, rats' behavior resembled that seen in prior experiments (Supplementary Tables 5–6). On the final test, we replicated retrieval-induced forgetting when we tested the competitor in the practiced arena (arena 1): rats recognized competitors more poorly than in the time or interference control conditions (Fig. 3c, left panel, arena 1 and Supplementary Table 6, ***$p_{(ANOVA)} = 0.0001$, $F_{(2, 20)} = 14.26$, ***$p < 0.001$, Bonferroni post hoc comparisons after a repeated-measures one-way ANOVA). This finding confirms that encoding an object across multiple arenas did not affect the forgetting phenomenon. Critically, however, rats also forgot the competitor when tested in arena 2 (Fig. 3c, left panel, arena 2 and Supplementary Table 6, $p_{(ANOVA)} < 0.0001$, $F_{(2, 22)} = 27.37$, ***$p < 0.001$, Bonferroni comparisons after a repeated-measures one-way ANOVA). The amount of forgetting was similar across test arenas (context × condition interaction, $p_{(int)} = 0.99$, $F_{(2,42)} = 0.008$, two-way repeated-measures ANOVA), showing that the effect generalized over contexts. In a control experiment (Experiment 4, Fig. 3c, right panel and Supplementary Tables 7–8), we also found that retrieval practice in arena 1 only harmed memory for the competitor in arena 2, and not other objects only seen in arena 2 (e.g., object F); thus, inhibition selectively acted on object memories that competed during retrieval practice.

Together, these findings indicate that retrieval practice inhibited competing memories in a cue-independent manner. This pattern is theoretically important because it shows that retrieval practice did not simply disrupt associations between the

practiced arena and the competitor, excluding purely associative accounts of the impairment, as in human studies[21,23].

**Durability of adaptive forgetting**. If adaptive forgetting processes cause lasting memory loss, forgetting should endure beyond the retrieval practice session. Experiment 5 replicated Experiment 1 with a 24-h delay before testing (Fig. 4a). The forgetting effect was significant and seemingly undiminished (Fig. 4a and Supplementary Table 9). Rats recognized competitors more poorly than objects in the time or interference control conditions (****$p_{(ANOVA)} < 0.0001$, $F_{(2, 21)} = 19.45$; ***$p < 0.001$ and ****$p < 0.0001$, Bonferroni comparisons after a one-way repeated-measures ANOVA, $n = 8$ rats). Thus, whereas retrieving a memory several times preserved its accessibility, it caused enduring forgetting of competitors, adapting memory to recurring patterns in its use. This forgetting persists beyond the initial transitory inhibition of competitors thought to driven by the prefrontal cortex.

**Role of the prefrontal cortex in driving adaptive forgetting**. If the adaptive forgetting process in Experiments 1–5 parallels retrieval-induced forgetting in humans, forgetting should require the prefrontal cortex. Evidence that the prefrontal cortex drives this effect in humans comes from several sources. For example, during selective retrieval practice, dividing participants' attention[42] or applying transcranial direct current stimulation to the right dorsolateral prefrontal cortex[27] undermines the control process, abolishing the forgetting effect. In imaging studies, selective retrieval engages lateral prefrontal cortex activation which predicts forgetting[24–26], with both activation and the forgetting effect linked to genetic variation in prefrontal dopamine[26]. Prefrontal control processes may disrupt competitors by inhibiting competing memories in the hippocampus or neocortex[25]. The rodent mPFC has been associated with attentional and inhibitory control processes[12–15], and has been proposed to resolve mnemonic interference via retrieval-related inhibitory processes that interact with hippocampal traces[39,40]. If so, disrupting the mPFC during retrieval practice may undermine inhibitory control and abolish retrieval-induced forgetting (Fig. 4b)[40].

Experiment 6 tested this prediction by reversibly inactivating the mPFC during retrieval practice. We injected the γ-aminobutyric acid agonist muscimol (0.1 mg/ml) into the mPFC bilaterally (Fig. 4c, d) 15 min before retrieval practice, and at the same point in the interference and time control conditions. We tested rats twice in each condition (retrieval practice, interference control, or time control), once with saline and once with muscimol. Because muscimol lesions can increase hyperactivity and perseveration, we waited until the next day to conduct the final test to allow the muscimol to wash out, ensuring that the drug did not affect performance. Although these side effects still could have affected retrieval practice, infusing animals with saline or muscimol did not alter exploration times during this phase (Supplementary Tables 10–11). In both the saline and muscimol conditions, rats preferred the novel objects during practice trials, indicating retrieval of the practiced object (Supplementary Figure 2, *$p < 0.05$ and ***$p < 0.001$, paired $t$ test).

After retrieval practice, control rats recognized competitors more poorly, compared with the interference or time control conditions (Fig. 4d, *$p < 0.05$, $t = 2.89$ and $t = 2.88$ respectively, Bonferroni comparisons). Critically, however, rats injected with muscimol showed less retrieval-induced forgetting than vehicle rats (interaction of drug with condition, $p = 0.04$, $F_{(2,29)} = 3.429$, repeated-measures two-way ANOVA). Bonferroni-corrected comparisons confirmed that rats' memory for competitors was worse when injected with vehicle than

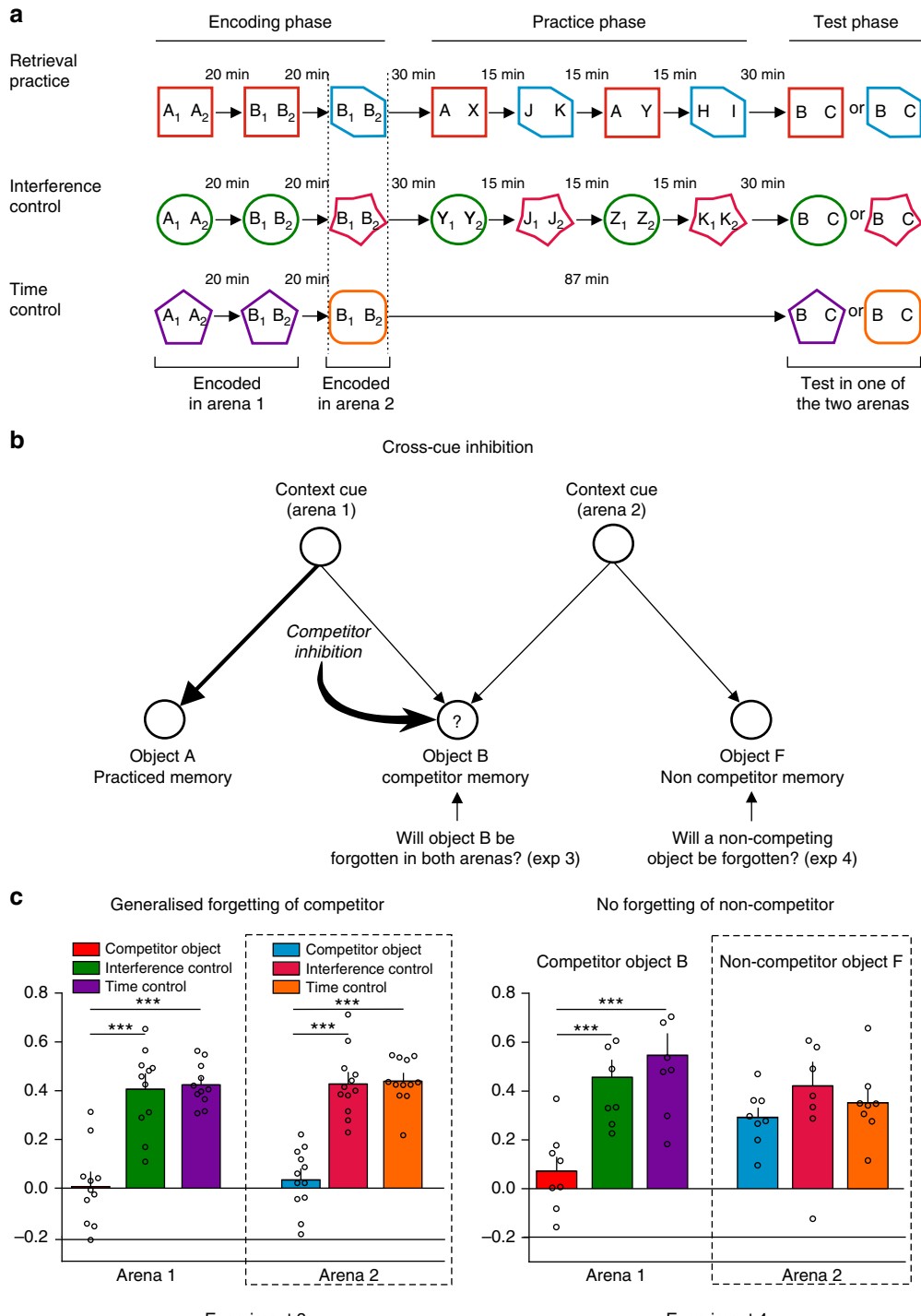

**Fig. 3** Forgetting is independent of the retrieval cue. **a** Experiments 3 and 4 include an additional session during encoding in which object B is also presented in arena context 2. During practice, we also interleaved sessions in arena context 2 to match the amount of additional exposure to the arenas. The final test was done in arena context 1 or 2. **b** The assumed retrieval dynamics during practice. Efforts to retrieve object A in arena 1 should trigger control processes to inhibit B, but not a non-competitor object F presented only in arena context 2. Object F is only included in Experiment 4 (additional encoding phase for F is not represented in **a**, but see Supplementary Figure 5B). **c** (Left) Discrimination index means ± SEM during test in arena 1 (left) or arena 2 (right). Individual values used to calculate the mean and SEM are presented as dots. Arena 1: ***$p = 0.0004$ ($t_{20} = 4.52$), $d = 1.96$ (RP vs. IC); $p = 0.0003$ ($t_{20} = 4.72$), $d = 2.04$ (RP vs. TC). Arena 2: ****$p < 0.0001$ ($t_{22} = 6.31$), $d = 2.36$ (RP vs. IC); ****$p < 0.0001$ ($t_{22} = 6.49$), $d = 2.85$ (RP vs. TC). Bonferroni post hoc comparisons after a repeated-measures ANOVA, $n = 11$ animals (arena 1) and $n = 12$ animals (arena 2). (Right) Discrimination index means ± SEM during test in arena 1 (left) or arena 2 (right). We found significant forgetting of B when tested in either arena context (cue-independent forgetting), but no forgetting of F. Arena 1: ***$p = 0.001$ ($t_{14} = 4.17$), $d = 3.02$ (RP vs. IC); ***$p = 0.0004$ ($t_{14} = 5.16$), $d = 2.22$ (RP vs. TC), Bonferroni post hoc comparisons, $n = 8$ animals. Arena 2: $p = 0.53$, $F_{(2,14)} = 0.65$, one-way ANOVA

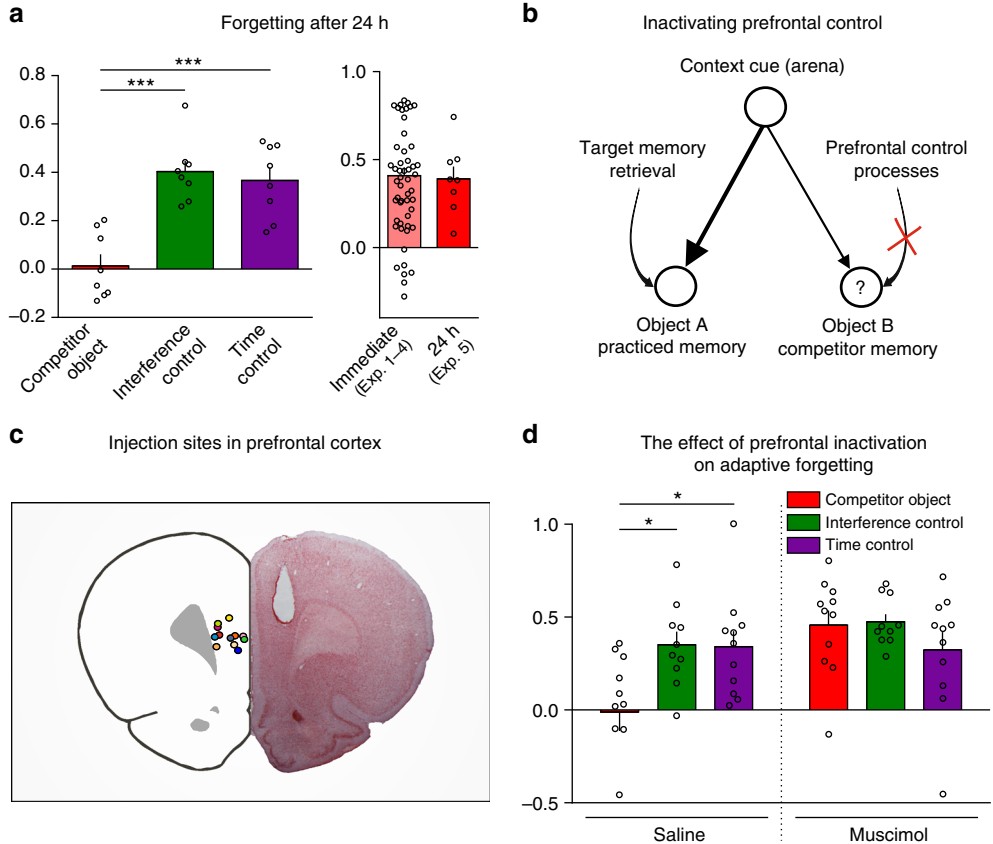

**Fig. 4** Forgetting persists after 24 h and depends on prefrontal control processes. **a** (Left) Discrimination index means ± SEM during test performed 24 h after the end of practice in Experiment 5. ***$p = 0.0001$ ($t_{21} = 5.64$), $d = 2.85$ (RP vs. IC) and ****$p < 0.0001$ ($t_{21} = 5.12$), $d = 2.59$ (RP vs. TC), Bonferroni post hoc comparisons after repeated-measures one-way ANOVA, $n = 8$ animals. Individual values used to calculate the mean and SEM are presented as dots. (Right) Forgetting effect (expressed as the difference between interference control and competitor discrimination index) after 24 h compared to that observed on the immediate test in Experiments 1–4. **b** Inhibition dynamics normally assumed to impair the retention of object B should be disrupted by blockade of prefrontal control processes during practice. **c** Examples of 12 injection sites in the prefrontal cortex for the retrieval practice condition (left). Safranin-stained section showing an example of the cannula track reaching the prefrontal cortex (right). **d** Discrimination index means ± SEM during test for saline-injected or muscimol-injected rats. We found a significant interaction drug × condition. Muscimol impaired forgetting of the competitor object. *$p = 0.022$ ($t_{29} = 2.89$), $d = 1.07$ (RP vs. IC) and *$p = 0.022$ ($t_{29} = 2.88$), $d = 1.04$ (RP vs. TC). Bonferroni post hoc comparisons after a two-way repeated-measures ANOVA. Competitor object, $n = 11$ animals; interference control, $n = 10$ animals; time control $n = 11$ animals

muscimol (**$p < 0.01$, $t_{29} = 3.47$). Indeed, injecting muscimol abolished retrieval-induced forgetting (Fig. 4d and Supplementary Table 12). This finding suggests that inhibitory processes mediated by the mPFC resolve competition not only between responses[12–15], but also between memories. Triggering lasting adaptive forgetting in rodents therefore requires mPFC, similar to the role of lateral PFC in humans.

**Adaptive benefits of forgetting.** In humans, inhibiting competing memories reduces the later need to control interference via the prefrontal cortex. In functional magnetic resonance imaging studies, these reduced demands on control are reflected in the decline in prefrontal activation when people retrieve the same memory several times, and the relationship of this decline to successful forgetting of competitors[24–26]. This inhibition of competing memories may affect traces in the hippocampus[24]. Given the mPFC's necessary role in causing retrieval-induced forgetting (Experiment 5), similar adaptive benefits of forgetting might arise in this species. If so, mPFC involvement may decline as rats repeatedly retrieve a memory because early trials enduringly disrupt competitors.

To test the adaptive nature of forgetting, Experiment 7 measured mPFC engagement at different stages of retrieval

practice with cFos immunostaining, a method often used to measure neural activity[43]. We ensured that cFos staining was truly due to retrieval practice by waiting 24 h between the encoding sessions and retrieval practice, allowing cFos expression from encoding to return to basal levels. Generally, cFos protein expression related to neural activity can be detected 90 min after behavioral testing[44]. Thus, to compare mPFC activity after the first 2 practice trials with activity after the third, we measured cFos expression at different times (Fig. 5). We evaluated one group 90 min after the second retrieval practice trial (Early); the other, 90 min after the third (Late). To ensure that only the Early or the Late retrieval practice trials affected their respective cFos measurements, we altered the intervals between trials so that we could clearly attribute cFos to these trials (Fig. 5 and Supplementary Methods). We made similarly timed measurements in the retrieval practice, interference control, and time control conditions (Fig. 5a, b).

Modifying the timing of the phases did not alter exploration (Supplementary Tables 13–14) or retrieval-induced forgetting (Fig. 5c and Supplementary Table 14). Critically, cell counts in mPFC revealed greater cFos immunolabeling in retrieval practice animals 90 min after the second practice (Early) compared to 90 min after the third practice (Late), and compared to all

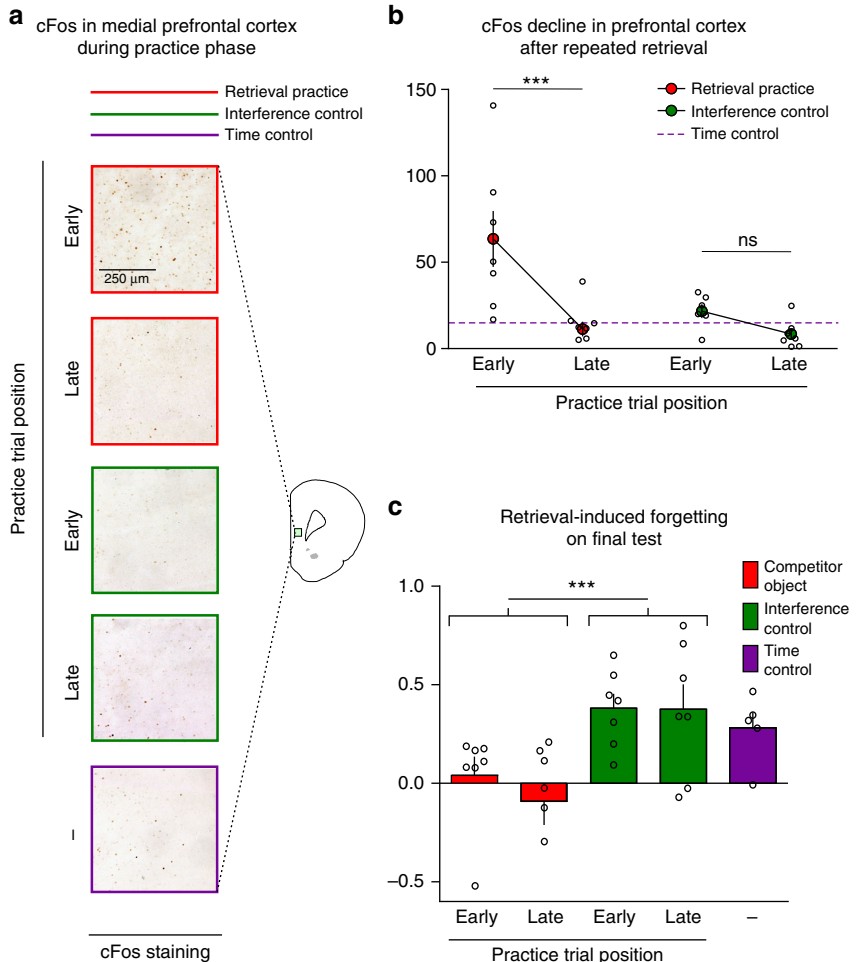

**Fig. 5** Prefrontal involvement declines after retrieval competition is resolved. **a** Representative images of cFos in the prefrontal cortex 90 min after the second practice trial (Early) or 90 min after the third practice trial (Late). Matched interference control groups were included and a time control group that remained in its home cage. **b** Number of cFos$^+$ cells per mm$^2$ in the prefrontal cortex expressed as means ± SEM for the Early and Late groups in the retrieval practice and interference control conditions (dotted line reports time control). Individual values used to calculate the mean and SEM are presented as dots. cFos was induced in the retrieval practice condition only in the Early group when competition needed to be resolved. ***$p = 0.0008$ ($t_{23} = 4.16$), $d = 1.66$ (Early vs. Late), Bonferroni post hoc comparisons, two-way ANOVA. **c** Discrimination index means ± SEM during test. Both Early and Late groups show forgetting. ***$p = 0.0006$ ($t_{26} = 3.89$), $d = 1.42$, unpaired $t$ test. Retrieval practice early and late, $n = 7$ animals each; interference control early, $n = 6$ animals; interference control late $n = 7$ animals; time control, $n = 6$ animals. NS not significant

controls (Fig. 5a, b, $F_{(4, 27)} = 7.41$, $p_{(ANOVA)} = 0.0004$, one-way ANOVA). Strikingly, after the third practice, activation did not differ from the control groups (Fig. 5b). We found a Practice Trial Position × condition interaction ($F_{(1, 23)} = 4,615$, $p_{(int)} = 0.04$, two-way ANOVA). Post hoc contrasts revealed that Early and Late treatments for the retrieval practice condition significantly differed, but did not for the interference control (Fig. 5b, ***$p < 0.0001$, $t_{23} = 4.16$, Bonferroni comparisons).

We considered whether mPFC activity during the final retrieval practice trial and the test affected cell counts from our Early condition, because both intervened before we measured cFos. This possibility is unlikely, however, because the Late condition itself exhibits little cFos activity. cFos in the Late condition estimates how much the third retrieval practice and test could have contributed to the Early condition because it isolates these phases' contributions: it measures cFos following the final test, 90 min after the third retrieval practice, and residual cFos from earlier practice trials should be minor given the protracted delays[45–47]. Critically, cFos counts in the Late and time control conditions were similar, even though practice was absent in the latter. This lack of cFos in the Late condition suggests that the

final practice trial and test did not influence the Early condition. This finding argues that cFos is high in the Early condition, returning to basal levels 180 min after practice, consistent with selective engagement of mPFC by early practice trials.

The contrasting patterns across the retrieval practice and interference control conditions are theoretically important. The lack of activity during the interference control suggests that forming episodic associations between the arena and novel objects does not engage the mPFC. Moreover, the lack of activity during the later retrieval practice trials suggests that retrieval per se does not engage the mPFC. Rather, the mPFC was engaged only when rats faced competition during retrieval (Early trials), consistent with a role in resolving interference. These findings suggest that actively forgetting competing memories during early retrieval practice trials reduces demands on prefrontal control during later trials, as in humans, consistent with an adaptive forgetting process.

## Discussion

Our findings point to a control mechanism triggered by retrieval that initiates adaptive forgetting in the mammalian brain. When

rats retrieved memories of an object, their later retention of other objects seen in the same context suffered. Rats recognized competing objects more poorly than objects in control conditions that replaced retrieval practice with rest or with novel object encoding, showing that forgetting caused by retrieval exceeded retention loss from time or interference. Retrieval-induced forgetting occurred on immediate tests and also after 24 h, without apparent reduction. Critically, this phenomenon was driven by rats' innate preference to explore novelty and the cognitive processes entrained by this adaptive behavior. The size and consistency of the forgetting were arresting: retrieving memories a few times nearly abolished recognition for competing objects, and this effect occurred in 88% of the 63 rats in Experiments 1–5. Thus, remembering caused forgetting in rodents as in humans[16–28].

Retrieval-induced forgetting in rats resembles the mechanism in humans. Retrieval-induced forgetting in humans occurs when competing memories impede recall, triggering inhibitory control processes thought to disrupt those competitors[16–28]. The functional and neural profiles in the two species exhibit parallels consistent with the involvement of inhibition. In humans, memories causing no competition during retrieval practice (e.g., memories encoded after retrieval practice or weakly encoded memories)[16,20,21,41] show no retrieval-induced forgetting, as with rodents: the same retrieval practice trials that impaired competitors encoded before retrieval practice did not affect those encoded afterwards (Experiment 2). In humans, retrieval-induced forgetting on delayed tests generalizes to multiple cues[18,20,21,23]. Rats' forgetting also shows this cue-independence, generalizing to the same object encountered across different contexts. Competition-dependence and cue-independence are hallmarks of inhibitory control in retrieval-induced forgetting[20,23]. Neurally, during retrieval practice prefrontal engagement predicts retrieval-induced forgetting in humans[24–26], whereas inactivation of this structure disrupts forgetting[27]. Similarly, reversible chemical lesions to the mPFC before retrieval practice abolished retrieval-induced forgetting. Strikingly, cFos imaging indicated a decline in mPFC engagement over retrieval practice trials, paralleling the initial engagement and then decline in lateral prefrontal activity in humans[24–26]. Declining prefrontal involvement across repetitions has been tied to greater forgetting of competitors, and thus reduced demands on prefrontally mediated control over hippocampal traces[24–26]. mPFC involvement suggests that active control mechanisms related to those that promote behavioral flexibility underlie retrieval-induced forgetting, consistent with this structure's role in mnemonic control[24–27,39,40,48–50]. Critically, the relationship between forgetting and declining prefrontal activity over repeated retrievals highlights how forgetting adapts memory to foster retrieval efficiency[24–26,51]. Together, these functional and neural parallels are consistent with the hypothesized species-general adaptive forgetting mechanism, although this bears careful scrutiny. Further work must establish the breadth of the mechanism, across species, whether species-specific variations exist, and potential adaptive benefits forgetting confers in the species' natural environments.

Equally striking as the robust evidence for retrieval-induced forgetting was the lack of interference effects on recognition. Final object recognition in the interference condition was unharmed (relative to the time control) by encoding three new objects in the same context as the critical item. The number of object memories associated to the arena was thus unrelated to recognition accuracy, even with a large sample. Relatedly, in Experiment 2, strengthening an object's association to the arena through retrieval practice, by itself, did not cause forgetting: when competitors were encoded after retrieval practice, rats' later recognition of them was unimpaired even though strong practiced memories could have interfered with their retrieval. Thus, having a single strong practiced competitor also did not impair retention. The lack of interference effects may have arisen from our final recognition test, which seems insensitive to such effects[4]. The absence of interference effects contrasts with the robust evidence for retrieval-induced forgetting and renders the latter unlikely to reflect historically emphasized passive forgetting processes.

Our findings are not alone in suggesting that retrieval-induced forgetting occurs in rodents. Converging evidence comes from a procedure in which rats learned to associate odors with food rewards[40]. After conditioning, additional training on a subset of associations impaired later memory for the remainder, compared to rats receiving no additional training. Poorer memory for competing odors was established by measuring rodents' tendency to dig for food in a cup of sand scented with a rewarded odor, over another scented with a non-rewarded odor: rats showed reduced preference for digging in the previously rewarded cup. mPFC lesions abolished this forgetting effect, as did lesions to the hippocampus. This work did not, however, rule out interference mechanisms as we have, nor did it establish functional parallels with retrieval-induced forgetting in humans. Moreover, its reliance on appetitive conditioning[52] and repeated training complicates comparisons to the human episodic memory phenomenon. As such, the case for active inhibition of competing episodic memories is less clear. Nevertheless, the similarities between this work and the current findings suggest that the processes observed here may generalize beyond our object recognition design.

The current findings may be related to a phenomenon that historically has not been viewed in terms of episodic forgetting: retrospective revaluation[53,54]. Retrospective revaluation resembles retrieval-induced forgetting procedurally. Rats first are trained that a compound stimulus (e.g., a light and tone) predicts reward, and, in a second phase, one stimulus is extinguished (e.g., the tone). Interestingly, extinction of one element alters conditioned responding to the non-practiced element, increasing responses to it. Thus, selective re-exposure to a subset of material affects omitted content, as with retrieval-induced forgetting. Associative models of retrospective revaluation emphasize the role of conditioned stimuli as reward predictors: changing the predictive status of one element by extinction causes revaluation of the competing predictor (light). The current work, however, highlights how retrieval processes may contribute. For example, re-exposures during extinction may engage retrieval processes that alter memory for the second element. However, whereas retrospective revaluation relies on strong associations between the compound's elements[54], retrieval-induced forgetting decreases under these conditions[37,38], because associations between targets and competitors eliminate competition. Relationships between the phenomena require further study.

The cellular and molecular mechanisms underlying the forgetting observed here remain unknown. One possibility is that suppressing competitors via prefrontal control triggers active forgetting mechanisms identified in molecular neuroscience. The small G protein RAC1, which is critical to active forgetting of conditioned fear in *Drosophila*, is up-regulated to hasten forgetting of outdated responses in reversal learning and interference procedures[2,11]. Moreover, dorsal hippocampal activation of RAC1 in mice also regulates time-based and interference-based forgetting in object recognition tasks similar to ours[55]. RAC1 activity is thus a viable mechanism underlying regulated episodic forgetting in mammals. Given these findings, we speculate that retrieval engages mPFC to induce episodic forgetting of competing memories via fronto-hippocampal inhibitory control inputs that signal the competitor's irrelevance; these suppressive inputs may, in turn, trigger RAC1 activation in the dorsal hippocampus. The pathway underlying this hypothesized inhibitory control/RAC1 coupling remains unclear, but may involve

GABAergic interneurons within the hippocampus[56]. The current procedure provides a flexible tool for evaluating this theoretical hypothesis in a species-general manner because the instinctive novelty preference capitalized on here arises in diverse species, including mice, guinea pigs, dogs, cats, horses, monkeys, zebrafish, ravens, and even in human infants[31,33,34]. Retrieval-induced forgetting, first established in humans[16,20,25], may constitute a broad model of adaptive forgetting that provides a critical link between high-level behavioral control and molecular mechanisms that drive forgetting at the synaptic level, showing how both prefrontal mechanisms and active forgetting work symbiotically to ensure flexible, adaptive behavior.

Over the past several decades, neurobiological research on memory has focused on mechanisms underlying memory storage[57–59]. The current work underscores, however, that the mnemonic fate of an experience is not determined solely by its encoding or consolidation, or by passive loss of plasticity; rather, it is governed by how organisms interact with memories to support behavioral goals. Here we establish adaptive forgetting as one powerful consequence of this interaction. We found that an evolutionarily selected motivated behavior, the drive to explore novel environmental features, entrained episodic retrieval in service of behavioral control, and, in doing so, tethered the shaping on the mnemonic ecosystem to organisms' adaptive needs. The striking size, consistency, and durability of this effect, its generality across species, and its characteristics suggest a foundational forgetting process. More broadly, these findings question whether forgetting is intrinsically problematic; rather, forgetting is a function supported by active mechanisms that help retrieve particular memories, a need that likely prevailed throughout the evolutionary history of memory—a "mental power"[1] possessing enough value to be conserved across mammalian species.

## Methods

**Animals**. One hundred and twenty-nine male adult Wistar rats between 2 and 3 months of age (weight 180–250 g) were group-housed (5 per cage), maintained in a 12 h light/dark cycle (lights at 7:00 a.m.) and constant temperature of 23 °C with ad libitum access to food and water. A different cohort was used for each experiment, taking place during the light phase of the cycle. The experimental protocols followed guidelines of the National Institute of Health Guide for the Care and Use of Laboratory Animals. The number of animals used is stated for each experiment (see Supplementary Methods). Sample size was calculated with G*power software (Universität Düsseldorf) based on an effect size from pilot studies.

**Apparatus**. Due to our within-subjects behavioral designs, all animals were exposed to a total of five arenas. Arenas were assigned pseudo-randomly to each phase, with the exception of arenas 5 and 7 that were used for habituation to objects presented as contextually novel during the practice phase. Animals that participated in Experiments 3, 4, 6, and 7 went through a shaping phase (see explanation in Supplementary Methods) and then exposed to two additional arenas. See Supplementary Methods for the descriptions of behavioral arenas and objects.

**Memory test for retrieval-induced forgetting**. We modified the spontaneous object recognition task to study retrieval-induced forgetting. We used a within-subject design for Experiments 1-5, a mixed within-subject (drug) and between subject (condition) for Experiment 6 and a between subject design for Experiment 7. Thus, each animal was subjected to each of the three conditions. The order in which they were exposed to each condition was pseudo-randomly assigned. Each experiment (with the three conditions) was conducted over a span of 3 weeks and each condition was separated from the next by 3 days. We conducted seven experiments: (1) to determine whether RIF can be manifested in rats, (2) show that the effects observed in our first experiment only arise for memories that compete during retrieval practice process and are not produced by associative blocking during the final test, (3 and 4) to determine that the RIF effect was independent of the context retrieval cue used during the final test, (5) to evaluate the durability of the effect over a 24-h interval, (6) to evaluate the involvement of the mPFC, and (7) to evaluate the adaptive nature of the RIF process. See Supplementary Methods for the training and retrieval protocols.

**Statistical analysis**. As a general rule, absolute exploration times during encoding were analyzed using paired $t$ test and absolute exploration times of the practice phase were analyzed using one-tailed paired $t$ test. Discrimination indexes during the test phase were analyzed using one-sample $t$ test against "0" as hypothetical value to determine significant memory retention. For Experiment 1, discrimination indexes were compared using a repeated-measures one-way ANOVA followed by Bonferroni post hoc comparisons (Fig. 1c, left). Absolute exploration times during test were analyzed using a paired $t$ test (Fig. 1c, right). Discrimination indexes of Experiment 2 (Fig. 2c) were analyzed using repeated-measures two-way ANOVA followed by Bonferroni post hoc comparisons. Discrimination indexes of Experiment 3 (Fig. 3c, left) were analyzed using a one-way repeated-measures ANOVA followed by Bonferroni post hoc comparisons for arena 1 and arena 2 conditions separately, as different groups of animals were used for each arena. Experiment 4 (Fig. 3c, right) was analyzed using a two-way repeated-measures ANOVA followed by Bonferroni post hoc comparisons. Experiment 5 was analyzed using a one-way repeated-measures ANOVA followed by Bonferroni post hoc comparisons. Experiment 6 was analyzed using a two-way repeated-measures ANOVA followed by Bonferroni post hoc comparisons with "drug" as a repeated measure. Cell counts from Experiment 7 (Fig. 5b) were analyzed using a two-way ANOVA followed by Bonferroni post hoc comparisons. Discrimination indexes depicted in Fig. 5c were compared using an unpaired $t$ test (competitor object vs. interference control).

## Data availability
All relevant data are available from the authors upon request.

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

## Acknowledgements

We thank Juan Facundo Morici and Magdalena Miranda for help with some of the experiments, John Duncan, James Rowe, Justin Hulbert, and Jorge Medina for useful comments on the manuscript, and Arthur Shimamura for early inspiration on this topic. This work was supported by research grants from the UK Medical Research Council (grant MC-A060-5PR00 awarded to M.C.A.) and the National Agency of Scientific and Technological Promotion of Argentina (ANPCyT) to N.V.W. (PICT 2008-00072 and PICT 2008-1065) and IBRO Return Home Fellowship 2012 to P.B.

## Author contributions

Conceptualization, M.C.A., P.B., and N.V.W.; methodology, M.C.A., P.B., and N.V.W.; investigation, P.B., N.V.W., F.G., and M.R; writing original draft, M.C.A., P.B., and N.V.W; writing review and editing, M.C.A., P.B., N.V.W., and F.G.; funding acquisition, P.B., N.V.W., and M.C.A.; resources, P.B. and N.V.W.

## Additional information

**Competing interests:** The authors declare no competing interests.

