## [Peer Review File · Nature Communications]

Reviewers' Comments:

Reviewer #1:

Remarks to the Author:

The manuscript by Bekinschtein et al. explores the phenomenon they refer to as “adaptive forgetting” and a role for the medial prefrontal cortex in this phenomenon in rats. The authors first show that rats show “adaptive forgetting” in a novel object recognition procedure. Specifically, rats are first exposed to two different objects in one context across two sessions. Then, rats are exposed to one of these objects and another novel object in the same context. Finally, rats are tested with the object that has not been seen since the beginning sessions. They find that rats “adaptively forget” this stimulus as indexed by their increased willingness to explore this object during this final test. The authors also perform a number of control experiments to show that these effects generalize across contexts and are unaffected by an “interference” phase before the retrieval tests. This shows that their effect is not the result of a retrieval failure as some might argue (e.g. Miller comparator hypothesis) and instead show that this effect is a result of a weakening of the original object memory. I commend the authors for the sophistication of their behavioural procedures and their attempt to specify the processes underlying their effect with their control conditions they use and the additional control experiments they run. I think these are an interesting set of data. However, I am genuinely torn about my recommendation for publication. I detail the reasons for this below.

My first and most important point concerns the wording used to describe these effects which makes the data seem more novel than they are. To me, it is very clear that the authors are showing a well-known effect called retrospective revaluation in a novel setting. Retrospective revaluation refers to the procedure whereby a compound stimulus (e.g. light + tone) is paired with reward. Then one of these stimuli (e.g. light) is presented alone and without reward. Interestingly, subjects will now judge the tone to be a better predictor of the reward and will increase their judgement of causality to this stimulus. The reverse is also seen. If you train subjects that the light + tone predicts reward, and then present the light with reward, subjects will now judge the tone less likely to produce reward. So the effect shown here then is in accordance with retrospective revaluation. In fact, the medial prefrontal cortex has been implicated in this effect before, though I think the manipulation here is considerably more sophisticated. So I think referring to this procedure as “adaptive forgetting” is somewhat misleading and probably not an accurate description of what is happening from an associative perspective given the data presented in the manuscript.

To my knowledge, this is the first time showing this effect in novel object recognition where subjects are not receiving a “reward” per se (though one might argue that the rats do find exploring novel objects rewarding). The strength of this manuscript then, comes down to what their particular experiments tells us about the associative nature of the retrospective revaluation effect and the role of the medial prefrontal cortex in this effect. I think there are a few important aspects of this paper from this perspective. But in some senses the novelty in these experiments also open up a number of questions about the nature of this effect that I am not sure are answered in the manuscript. For example, why does the interference control not produce the same effect? That is, why is it supposed that rats associate stimuli A and B together within the same context but not another cues which are also subsequently presented in these contexts in the “practice phase”? Is it just because rats experience objects A and B first and 15 minutes closer together than the other objects which occur in a block? In the manuscript there is a suggestion that rats may experience object A or B first (i.e. p4, line 117)? Is the description of the protocol in the figures just an example of one set of counterbalancing? If so, can we see the data on conditions where B is given first, and then where A is given first? Unless I am missing something from the argument, it is not entirely clear to me why the interference control does not produce the same effect and specifically why objects A and B are associated together. Usually, rats would be exposed to stimuli together to produce these effects (i.e. object A and B in arena 1 at the same time). Would the authors predict that object A alone in arena 1 during the practice phase would be

enough to induce the forgetting of B? Or does it need to be compared with another stimulus to be "retrieved"?

The experiment testing for the context generalizability of the forgetting effect is interesting (though it suffers from the issue I describe above also). Specifically, the authors find that the "adaptive forgetting" effect generalizes across contexts. This suggests that the rats really do "forget" the object. That is, these results cannot be described in terms of a simple reduction in context-object association because if this were the case then the effect would not occur in a new context. The critical control for this is the interference control, where rats just exposed to other objects in the same context which are not "related" to the critical forgotten object do not show forgetting of the critical object- as an aside I thought the authors might better make this point about the importance of the interference control in this particular experiment. This is an important point because it tells us something about the associative structure of what is going on. I thought the authors might better focus on this point and discuss the consequences that this experiment has for our understanding of the associative structure of their effect (and generally retrospective revaluation).

Another novel point of the manuscript is that the role of medial prefrontal cortex in this effect is transient. That is, cFos indicators of neural activation in medial prefrontal cortex in the adaptive forgetting effect occurs in the first encounters with the object during the practice phase but indicators of activity taken later in the practice phase do not elicit significant results. There is probably a more elegant way to have demonstrated this (e.g. muscimol infusion after two practice sessions have taken place), but the point is novel and important. It is consistent with other work looking at medial prefrontal cortex and the resolution of response competition (in particular, prelimbic cortex on goal-directed responding, attentional tasks etc.), where the role of prelimbic cortex is often transient. This again tells us something about the nature of the "adaptive forgetting"/retrospective revaluation effect. Sometimes the role of the prelimbic cortex is transient and sometimes it is not. For example, the prelimbic cortex is involved in both the acquisition and expression of context-dependent associations. So, the transient nature of the medial prefrontal cortex in this effect suggests early encounters with the competitor object elicit the inhibition effect that wanes across time. The authors again might spend more time talking about the significance of this point and what it tells us about the associative nature of the effect.

The final significant point I would raise is that the manuscript is very hard going. Indeed, it took me significantly longer than usual to review it. I found that the graphical representations of the data were quite confusing and I kept needing to go back and read the results to understand exactly what the authors had found. For example, in Figure 3C there is suddenly a reference to non-competitor F which is not referenced in the figure illustrating the design. Generally, I don't think the results are represented graphically in a straight-forward manner. Further, the wording describing the effects is very verbose and confusing. And the results section and description of the methods is entirely too long. I think the authors could significantly shorten these sections and focus on the main points of their data or else I am concerned that the main points of the data will not get across to their readers.

In summary, I think there are some important points that come out of the manuscript. If the manuscript goes to publication here I think the authors need to recognize the extensive associative literature that has come before this (i.e. retrospective revaluation) and the associated controversy related to the associative nature of this effect, and focus more on the unique points related to their data (i.e. context generalizability of their effect, and the transient role of the medial prefrontal cortex in this design). It is also very important that the authors make it clear why objects A and B by necessity become associated in a way that does not generalize to the other object in the same arena in the interference control condition?

Below is a (non-exhaustive) list of references which are related to the present manuscript and should be discussed and cited in the manuscript:

Dickinson, A. (1996). Within compound associations mediate the retrospective revaluation of causality judgements. *The Quarterly Journal of Experimental Psychology: Section B*, 49(1), 60-80.

Le Pelley, M. E., & McLaren, I. P. L. (2001). Retrospective revaluation in humans: Learning or memory?. *The Quarterly Journal of Experimental Psychology: Section B*, 54(4), 311-352.

Corlett, P. R., Aitken, M. R., Dickinson, A., Shanks, D. R., Honey, G. D., Honey, R. A., ... & Fletcher, P. C. (2004). Prediction error during retrospective revaluation of causal associations in humans: fMRI evidence in favor of an associative model of learning. *Neuron*, 44(5), 877-888.

Ghirlanda, S. (2005). Retrospective revaluation as simple associative learning. *Journal of Experimental Psychology: Animal Behavior Processes*, 31(1), 107.

San-Galli, A., Marchand, A. R., Decorte, L., & Di Scala, G. (2011). Retrospective revaluation and its neural circuit in rats. *Behavioural brain research*, 223(2), 262-270.

Tran-Tu-Yen, D. A., Marchand, A. R., Pape, J. R., Di Scala, G., & Coutureau, E. (2009). Transient role of the rat prelimbic cortex in goal-directed behaviour. *European Journal of Neuroscience*, 30(3), 464-471.

Sharpe, M. J., & Killcross, S. (2015). The prelimbic cortex uses higher-order cues to modulate both the acquisition and expression of conditioned fear. *Frontiers in systems neuroscience*, 8, 235.

Stout, S. C., & Miller, R. R. (2007). Sometimes-competing retrieval (SOCR): A formalization of the comparator hypothesis. *Psychological Review*, 114(3), 759.

Melchers, K., Lachnit, H., & Shanks, D. (2004). Within-compound associations in retrospective revaluation and in direct learning: A challenge for comparator theory. *The Quarterly Journal of Experimental Psychology Section B*, 57(1), 25-53.

Reviewer #2:

Remarks to the Author:

The authors report findings from a rodent model of adaptive forgetting, the retrieval-induced forgetting effect. This is an excellent paper that convincingly demonstrates that the retrieval of competitor memories is inhibited by retrieval practice of other memories from the same context, a phenomenon which has been extensively studied in human subjects but has received very little attention in animal models. Indeed, the notion that retrieval inhibition plays a key role in forgetting has not really penetrated the animal behavioral neuroscience of memory community, much to the detriment of the study of the neuroscience of memory in my opinion, so this paper has high potential impact.

The experimental design is very clever, taking advantage of spontaneous object investigation and the fact that rodents readily associate objects with the context where they were encountered. Importantly, the authors have carried out the appropriate controls to clearly rule out alternative explanations (i.e. other than retrieval-induced inhibition of the competitor memory), including time and interference controls, but also demonstration of cue independence, specificity to competitor memories and durability of the effect out to 24 hours. These controls are crucial: they not only rule out trivial (but unlikely) alternatives, but they really define the nature of the memory effect and distinguish it from a variety of other plausible memory phenomena. The inclusion of these controls is not surprising given that the senior author is the leading authority on this phenomenon as it has been studied in humans. Nevertheless, it is notable that this particular experimental design allowed the authors to include the relevant controls in a manner that kept other aspects of

the task comparable to the conditions that produced the basic retrieval-induced forgetting effect. Finally, these experimental procedures provide an excellent approach for studying these adaptive forgetting/retrieval inhibition effects using a variety of invasive manipulations that cannot be done with human subjects. The authors have already performed two of these, muscimol inactivation and measurement of IEG expression in the medial prefrontal cortex, and it is likely this paper will spur additional research.

I am generally enthusiastic about the paper, but I do have a few concerns which should be addressed.

A similar phenomenon was reported in rats by another laboratory (Wu et al, 2014, Hippocampus) and although the authors cited the paper, they should discuss it in more detail. The previous paper reported that retrieval inhibition occurred in an olfactory learning task and that medial prefrontal inactivation (and hippocampal inactivation) disrupted this effect. However, in my opinion the previous finding should not detract from the novelty of the present results. The Wu et al paper had a number of inadequacies, including that it used an instrumentally reinforced behavior (rather than a spontaneous behavior as in the present study), which could have unknown effects, and it was impossible to rule out alternative explanations for the finding. It did not include the critical controls that really define the phenomenon, as described above. For these reasons, I think the present study is not simply an extension of previous work, but a novel and much more convincing demonstration of retrieval inhibition in rodents. Nevertheless, the previous paper is, at minimum, consistent with the present results and the broader literature on retrieval inhibition. The authors should more explicitly discuss the previous findings and theoretical framework for mPFC inhibitory control, and how they relate to the present results.

Minor Concerns

The authors say on p 4, line 124 "... they had retrieved an episodic-like, place-specific memory ...". Why 'episodic-like'? This task doesn't seem to meet the typical 'what, where, when' requirements that other authors have suggested for 'episodic-like' memory (a phrase I find a bit too equivocal anyway, but others disagree). Also, 'place-specific' may be confusing if the authors actually mean context-specific, since rodents can learn to match objects to contexts and to specific locations within a context (i.e. 'places').

The authors sometimes use the term 'adaptive forgetting', sometimes 'inhibition', and sometimes 'retrieval-induced forgetting'. I have no objection to any of these, but I think it would be helpful to be clear and explicit about what each of those terms means and whether they're being used interchangeably, especially for readers who are not familiar with this literature.

mPFC lesions can cause hyperactivity and perseverative responding. Can the authors rule out the possibility that these non-mnemonic effects of the muscimol influenced their results?

The cFos data are interesting and the differences between early and later practice fit very nicely with the author's theoretical framework on the role of the PFC. However, although the peak cFos expression may occur at 90 min, I assume there is a gradient of expression surrounding that time point and I think most. It looks like all of the rats continued training and testing beyond the time point of interest for cFos expression. I would like to see a discussion of how these subsequent events might have contributed to their cFos measures.

The authors might consider putting task schematic illustrations in figures 4 and 5, like those in earlier figures. This would be particularly useful for the cFos study.

If practical, the authors might consider putting additional descriptive labels on bar graphs (e.g. B vs C and J vs D under the red bars in Fig 2C, etc.) in addition to the current text.

Reviewer #3:

Remarks to the Author:

In a series of experiments using the novel-object paradigm in rats, Bekinschtein and colleagues sought to test the hypothesis that repeated exposure to a familiar object in a familiar context will lead to retrieval-induced forgetting of a previously learned object in that same context due to a competition process that is mediated by the medial-prefrontal cortex (mPFC). They propose that retrieval-induced forgetting is the result of active inhibitory control. Their retrieval-induced forgetting hypothesis was supported in an elegantly designed Experiment 1, and controlled for in a series of other experiments, though the claims of an active inhibitory control mechanism, paralleling that found in humans, remain somewhat speculative.

While I found the main results of the paper quite interesting, I admittedly found it a bit of a struggle to read. As a general comment, though the methods were described in detail on pg 17, the multi-stage conditions for 7 experiments, each with multiple objects and contexts made it very difficult to follow the procedures. Having read the detailed methods section on pg. 17 prior to the results, I was working very hard to keep all the experiments, contexts, phases, trials, and timelines straight. The schematics of the experimental designs referred to in the results (Fig. 1A, 2A, 3A) help to make sense of it all. For readers who review the specific methods before the results, it will be of great use to refer to these study design schematics within the detailed methods section.

My primary reservations with the paper relate to the claims that the retrieval-induced forgetting is (1) an active inhibitory control mechanism mediated by the mPFC, and (2) that this is a species-general phenomenon.

Based on the methods used (observation of exploratory behavior in a novel-object paradigm, silencing of mPFC, and IEG expression of c-Fos in mPFC following retrieval), it is not clear how the findings represent 'active inhibitory control' over forgetting. The authors have nice, well controlled findings in Experiment 1 in which repeated exposure to Object-A following exposure to both A & B within a context results in less exploration of familiar/distractor Object-B at a later timepoint when tested in that same encoding/sampling context. Though the results of Experiment 6 (mPFC inhibition) and Experiment 7 (mPFC IEG expression) suggest that the mPFC plays a role in mediating this retrieval-induced forgetting, it remains a stretch to state that inhibitory control is the specific mechanism, as this was not explicitly tested. To support this claim, the authors should provide a direct measure of inhibitory control mechanisms in the present study.

One of the main objective of the study was to determine if the prefrontal cortex plays a conserved role across species in mediating active forgetting. In the mPFC muscimol experiment (Experiment 6), it is specified that rats were assigned to one condition (RP, IC, or TC) for each a saline and a muscimol test. This experiment did support the proposal that the retrieval-induced forgetting is mediated by mPFC, as silencing this region reduced forgetting of the distractor object during the test session. It was less clear what was gained by Experiment 7, in which expression of c-Fos was observed following retrieval of the highly familiar object, but, from what I could tell, mPFC c-Fos expression was not tested following the actual test phase, which would allow you to examine the activity of the mPFC at the time during which the retrieval-induced forgetting is observable.

Instead, the results of Experiment 7 show a decline in mPFC expression of c-Fos over repeated retrieval practice sessions of target Object-A, suggesting, in line with what is found in humans, that the mPFC becomes less active as an item becomes more familiar. The authors should provide some supporting evidence from other studies in rodents as well that support the weakened dependence on mPFC for familiar items. To support the argument that the mPFC is supporting retrieval-induced forgetting of the distractor item following repeated exposure to the target item, an extra c-Fos experiment should be added, where c-Fos expression in mPFC is analyzed following

retrieval of distractor Object-B in the test phase (as well as a comparison of c-Fos expression following retrieval of target object-A in the test phase). This will provide a more direct measure of mPFC activity accompanying forgetting of the distractor item. Also, the full statistics need to be reported for the ANOVA results here, including the main effects, degrees of freedom, error (reported for second, but not first F value), and effect sizes.

Additionally, in Experiment 7, the authors assessed c-Fos expression in mPFC following 2 (early) or 3 (late) retrieval sessions of the familiar item (Object-A), and found reduced c-Fos expression after late, relative to early, retrieval. Was the same retrieval timeline as described in Experiment 1 followed in Experiment 7 - specifically, was the 3rd retrieval practice trial (late) conducted only 15 min after the 2nd retrieval practice trial (early)? If so, would you not expect that the c-Fos expression from retrieval trial 2 would have been detectable, albeit at a lower expression level, 90 min after the 3rd trial, and this cumulative c-Fos protein expression across multiple retrieval events would limit your ability to determine the unique c-Fos expression induced by retrieval trial 3 vs. retrieval trials 2&3 in these animals? Given that the c-Fos levels were reported to be lower following the 3rd trial, these results would argue against this cumulative expression argument, but this potential confound should be addressed.

My second reservation with the study involves the heavy focus on drawing parallels between the findings in the present study, with analogous processes occurring in humans. Throughout the intro and results, the authors repeatedly state that their findings parallel retrieval-induced forgetting in humans. Similarly, the discussion emphasizes the parallels between the results of the present study and parallel findings in humans, suggesting that their findings of retrieval-induced forgetting is a 'species-general phenomenon'. While it is certainly valuable to make these connections across species, the authors refer largely to human literature in their discussion of their rodent results, with comparably little discussion of related findings within the rodent literature. The results of the present experiments need to be better supported by what is currently known (and unknown) about this process in the rodent brain before an extension to its applicability to humans and other species can be made.

Finally, related to the active forgetting hypothesis across species, in the introduction, the authors do describe the mechanisms of active forgetting in drosophila, involving endocytosis of GluA2/AMPA receptors, which is a cellular mechanism that may very well underlie the forgetting seen in the present study, and which would be possible to test (though I recognize that was not the goal of the present study). The point here being, that they have proposed a testable mechanism in one species, yet claimed that their results are parallel with a different mechanism in another species (active inhibitory control in humans). Given that these cellular mechanisms were not tested in the present study, it would be better to save the discussion of these potential cellular mechanisms for the discussion section, and to streamline the introduction and results to focus specifically on the results of their study (there are many interesting ones), and to discuss the results of their experiments in the context of other relevant findings in rodents. While, in the discussion, the authors do propose potential hippocampus-mediated inhibitory control cellular mechanism involving RAC-1 activity, they do not discuss potential cellular mechanisms within the mPFC, which is the focus of their studies. This should be addressed.

Specific comments:

Results - the full statistical results from the ANOVA and repeated measures ANOVAs including degrees of freedom, as well as effect sizes should be presented for each finding in each experiment (i.e. line 234), rather than just presenting the p value. Also, the exact p-values should be presented rather than i.e. $p < 0.05$. See related comment for ANOVAs in Experiment 7.

Figure: It may be a result of format changes when the files were converted for submission, but the figures were difficult to read (blurry legends), and the c-Fos staining was not visible. The resolution of the figures should be improved.

Minor points:

The methods should be revised for repetition - for example, it is mentioned multiple times that quantification of exploratory behavior was measured manually using hand chronometers (line 725, line 736, and in the description of each condition).

Typo line 705 - randomly

Typo line 750 - period missing after phase 2

Typos throughout referring to figures. Should be consistent. I.e. line 263-266 -- I.e. Fig. 2A, Fig 2B, f, Fig. 2 C, Fig. 2C can be all be found. Similar findings through paper.

Typos line 201 - use decimals rather than commas in t-test results.

Be consistent in the terminology used. Given the complexity of the design, it is often unclear if the terms are being used interchangeably to refer to the same type of object, or to a different type of object entirely.

- Familiar object, practiced object, studied object
- Novel object, unpracticed object
- Encoding phase, sampling phase
- arena, context

Response to Reviews:
Manuscript NCOMMS-18-02762

Reviewer 1

Reviewer Comment 1.1. The manuscript by Bekinschtein et al. explores the phenomenon they refer to as “adaptive forgetting” and a role for the medial prefrontal cortex in this phenomenon in rats. The authors first show that rats show “adaptive forgetting” in a novel object recognition procedure. Specifically, rats are first exposed to two different objects in one context across two sessions. Then, rats are exposed to one of these objects and another novel object in the same context. Finally, rats are tested with the object that has not been seen since the beginning sessions. They find that rats “adaptively forget” this stimulus as indexed by their increased willingness to explore this object during this final test. The authors also perform a number of control experiments to show that these effects generalize across contexts and are unaffected by an “interference” phase before the retrieval tests. This shows that their effect is not the result of a retrieval failure as some might argue (e.g. Miller comparator hypothesis) and instead show that this effect is a result of a weakening of the original object memory. I commend the authors for the sophistication of their behavioural procedures and their attempt to specify the processes underlying their effect with their control conditions they use and the additional control experiments they run. I think these are an interesting set of data. However, I am genuinely torn about my recommendation for publication. I detail the reasons for this below.

Author Response 1.1. We thank the reviewer for their very nice remarks about the work!

Reviewer Comment 1.2. My first and most important point concerns the wording used to describe these effects which makes the data seem more novel than they are. To me, it is very clear that the authors are showing a well-known effect called retrospective revaluation in a novel setting. Retrospective revaluation refers to the procedure whereby a compound stimulus (e.g. light + tone) is paired with reward. Then one of these stimuli (e.g. light) is presented alone and without reward. Interestingly, subjects will now judge the tone to be a better predictor of the reward and will increase their judgement of causality to this stimulus. The reverse is also seen. If you train subjects that the light + tone predicts reward, and then present the light with reward, subjects will now judge the tone less likely to produce reward. So the effect shown here then is in accordance with retrospective revaluation. In fact, the medial prefrontal cortex has been implicated in this effect before, though I think the manipulation here is considerably more sophisticated. So I think referring to this procedure as “adaptive forgetting” is somewhat misleading and probably not an accurate description of what is happening from an associative perspective given the data presented in the manuscript.

Author Response 1.2. We thank the reviewer for pointing us to the literature on retrospective revaluation, which is rich and informative. The possibility of linking our findings to this literature is inviting because it could expand the scope of the work to an important research domain. After reading a number of papers on the subject (several recommended by the reviewer and others that we found), this literature is clearly something that we will bear in mind, moving forward with the line of work we are reporting in this article.

After immersing ourselves in this work, however, we came to a different conclusion than the reviewer about the strength of the relationship between our work and retrospective revaluation: the relationships between the two phenomena are, at present, speculative, and in need of conceptual and empirical development. For these reasons, it would be premature to frame our

findings as tests of ideas in that literature. Indeed, we believe it would be inappropriate, as our ideas, designs, and controls originated from the human retrieval-induced forgetting literature, and such reframing would not be a truthful representation of the intentions of this work.

We elaborate on our reasons for this judgment below and we hope that the reviewer will see our point.

Relationships Between Current Findings and Retrospective Reevaluation (hereinafter, RR).

Similarities. It is clear why the reviewer drew comparisons to RR. First, RR, like our procedure, presents animals with two cues in an initial stage, and it then follows this with repeated presentation of one of the cues; finally, in a third stage, there is a test of the non-repeated cue. So, both phenomena look at the effects of an interpolated processing interval on a stimulus that was left out, as measured on a final phase. Structurally, and procedurally, the comparison is appropriate and it seems that there really could be an interesting relationship to be mined here. We are glad to know of this work.

Dissimilarities/Difficulties with the Comparison. Unfortunately, the array of differences in the procedures, properties, and findings is too large to ignore, rendering the comparison speculative and uncertain. Below, we highlight some of the differences that should lead someone to have honest doubts about the comparability of the procedures

1. *Spontaneous Object Recognition (hereinafter, SOR) is Different from Classical Conditioning.* RR paradigms are (when done in rodents) largely conditioning paradigms. A fundamental feature of such paradigms is the prediction of an Unconditioned Stimulus Event by a Conditioned Stimulus—indeed the major models of conditioning, and also of RR in particular, center around the issue of predicting a US event.

In SOR, a rat is simply placed in an arena to freely explore objects. There is no clearly identifiable Unconditioned Stimulus to predict. One could try to reframe the SOR in terms of the constructs of conditioning, but we believe that such reinterpretation is forced---it is not obvious. We are not saying that this is not possible or that the reviewer's proposal is incorrect, but rather that any such re-interpretation would be speculative, and in need of better justification that our data would allow. Instead, SOR task is more conventionally used to study episodic memory processes, as we have done, and a sizeable literature has grown linking this task to MTL contributions to memory.

2. *There is No Competition for Prediction in our RIF Paradigm.* RR paradigms pair a “target” and a “companion” stimulus concurrently in Phase 1, setting up a competition for their predictive status of the paired US (i.e. “is it the tone or the light that predicts the US?). In our rodent retrieval-induced forgetting paradigm, the two stimuli are presented in the arena at different times—20 minutes apart from one another. Indeed, the rodents first see two copies of stimulus A, followed 20 minutes later by two copies of stimulus B. They occur on entirely different occasions and are distinct events to the rat. We don't feel as though we could credibly justify the assumption that A & B are competing for prediction for a common US event, particularly given that (a) there are no US events in general, and (b) even if something could be construed as a US event, they would not be the same US event, but distinct occurrences on different occasions.

The independent presentation of the objects on two occasions is a critical difference, leading the analogy to break down. Consider the analogous circumstance in classical

conditioning. In a conventional classical conditioning paradigm, one can condition a tone to a shock and a light to a shock, if done on different occasions and rodents will learn those associations separately—they are two predictors of the US event. But if a tone and a light occur concurrently, it's unclear which one deserves credit for predicting the particular US event. This is the ambiguity that RR fundamentally addresses, and that is simply absent in our paradigm.

3. *Within-Compound Associations Cannot be Assumed.* The RR phenomenon relies on presentation of a compound stimulus, and on the formation of an association between the Target stimulus and the Companion stimulus. The effect increases as this association increases and theoretical accounts of the phenomenon often rely on this association (see, e.g., Miller & Witnauer, 2016¹, for a nice review). The association between A and B is intentionally facilitated by concurrent presentation of A & B, and long-duration CS presentations.

In contrast, our rodent RIF procedure presents objects on different occasions, separated by 20 minutes. It not obvious that this protocol would facilitate the formation of an A – B association. Indeed, this design choice—to separate the presentation of the A and B stimuli—was motivated by the consistent finding, from the retrieval-induced forgetting literature, that associations between the target and the competitor items reduce or eliminate the phenomenon (see Anderson, 2003² for a review). So, we were deliberately trying to reduce A/B associations, if possible. Of course, we cannot rule out that A and B are associated—they could be; but assuming that they *are* associated, given such vast differences in presentation timing would be imprudent at best—and would be rightly questioned by sceptical readers.

4. *Parametric Characteristics Differ Dramatically Between RR and RIF.* Although RR is a reliable phenomenon, it depends critically on adherence to certain design parameters (see Miller & Witnauer, 2016¹ for a review). One particularly important parameter concerns the number of repetitions in phase 2—i.e. the phase in which the companion stimulus is either deflated or trained. For RR to be reliably observed, a very large number of extinction trials is required on the companion stimulus. According to Miller & Witnauer:

“With respect to RR, although we often found that moderate numbers of extinction trials (e.g., 30) with the companion cue following cue competition treatment was sufficient to eliminate responding to the companion cue, a larger number of associative deflation trials with the companion stimulus was required to produce a robust RR effect. For example, Blasidell et al. (1999)³ found that 200 extinction trials of a blocking CS were inadequate to produce RR (recovery from blocking in this instance) with their parameters, but 800 trials yielded RR.”

Although we have found published incidences of RR with as few as 48 deflation trials (e.g., see San-Galli, Marchand, Decorte, & Di Scala, 2011)⁴, this is still vastly different than what we observed here. Our design produced exceptionally robust retrieval-induced forgetting in as few as 2-3 trials (akin to what produces RIF in humans). Given the very large difference in repetitions required to produce the effect, it seems quite likely that they are produced by different mechanisms. At a minimum, this disparity makes it imprudent to simply assume that they are the same phenomenon, and we do not feel that we could argue this credibly.

5. *RR is highly Context Dependent, whereas RIF is cue-independent* RR effects appear to be specific to the context in which RR treatment occurs (see Miller & Witnauer). That is,

if rats are tested in a context that differs from the one in which deflation occurs, RR is no longer observed. In direct contrast, our studies clearly demonstrate that RIF generalizes to a novel context in which retrieval practice was not performed. Here again, this argues that a different mechanism may underlie these phenomena, especially when considered with the very large array of important differences described above.

Synopsis.

These are just a few of the major difficulties with the comparison. In fact, there are more than this (e.g. other ways in which the pattern of findings in our studies contradict what would be expected based on RR), but we think that, taken collectively, the foregoing points are sufficient to prescribe caution, at a minimum, about whether the initial structural resemblance of the paradigms warrants the inference that they are the same. These are the considerations that led us to draw a different conclusion than the reviewer about the strength of the relationship between these phenomena.

Author Action Taken 1.2. Despite these significant doubts, we very much agree with the reviewer that the structural resemblance of the procedures warrants discussion of RR. We therefore included a new paragraph in the discussion, drawing readers' attention to the relationships of these procedures, and pointing out potential relationships and differences (5th paragraph in the discussion). In this paragraph, we argue, in fact, that retrieval-induced forgetting—rather than being explainable in terms of RR, instead may be a basic memory retrieval process that could contribute to and inform theories of RR. This seems to be a logical possibility that is worth considering by the members of the RR research community. Could the construct of retrieval-induced forgetting contribute to understanding this crucial phenomenon? We agree with the reviewer about the structural parallels in the paradigms, and we also believe that selective presentation of the companion stimulus in deflation phase could very well engage our retrieval-related inhibition processes, possibly contributing to the RR phenomenon. This could be a stimulating new direction.

One very important difference between the conditions that produce RR and the conditions that produce retrieval-induced forgetting is that the former rely on inter-stimulus associations, whereas the latter is eliminated by them, and can even be reversed (facilitation) in some cases. We highlighted this difference in our new paragraph as something to consider. However, we do not view this as evidence for why one should not consider the role of retrieval processes in RR. One way of viewing RR paradigms is as being on the very high end of the inter-stimulus association spectrum, which, when combined with massive extinction trials, provides very ample opportunity for the retrieval of the original event (both the target and companion together), which ought to enhance accessibility in memory, not inhibit it (there is evidence of the corresponding effect in the RIF literature). This suggests that if all semblances of inter-item associations are eliminated in an RR design, the inhibition dynamic may be in maximum force. The fact that RR is not observed under those conditions may actually reflect the difficulties rats may have in retrieving the target dimension, given that retrieval (extinction) of the companion has repeatedly suppressed it. So, in fact, RR may be limited by the dynamics of RIF.

As is apparent from the above, we have been quite stimulated by the reviewer's suggestion, and think it's worthwhile to pursue, even if came to some different conclusions than the reviewer.

Naturally, we retained our characterization of our findings in terms of adaptive forgetting. We did this because (a) it was our intention, from the start, to study RIF in rodents, and this intention led to all the specific designs we adopted; (b) the array of findings we have observed in rodents accords incredibly well with the known properties of RIF in humans (all predicted in advance, based on that literature), and (c) the use of the discrimination index in SOR is widely used in

rodent research on episodic memory and so is a reasonable measure of retention by which to index forgetting. Based on these considerations, we believe that we have an internally consistent, credible interpretation based on an extremely large and mature literature (500-1000 papers) that will drive new ideas and findings in the field.

Reviewer Comment 1.4. To my knowledge, this is the first time showing this effect in novel object recognition where subjects are not receiving a “reward” per se (though one might argue that the rats do find exploring novel objects rewarding). The strength of this manuscript then, comes down to what their particular experiments tells us about the associative nature of the retrospective revaluation effect and the role of the medial prefrontal cortex in this effect.

Author Response 1.4. We respectfully disagree with the reviewer about the foregoing synopsis. The work aligns extremely well with the sizeable literature on retrieval-induced forgetting and every aspect of the design choices and hypotheses originated from that human literature. It took considerable effort to take properties demonstrated over decades in the human literature and evaluate whether they are true in our adapted paradigm with rodents. The strength of this manuscript, therefore, has less to do with what it tells us about RR, which it was not intended to address, and more to do with the highly novel and potentially very important line of work it stands to initiate in rodent research on memory—a line of work that RR researchers may ultimately benefit from. We are grateful to the reviewer for their ideas about RR, but our paper is not about that phenomenon. We’d certainly welcome additional follow on work designed to evaluate the relationship between the two effects though.

Reviewer Comment 1.5. I think there are a few important aspects of this paper from this perspective. But in some senses the novelty in these experiments also open up a number of questions about the nature of this effect that I am not sure are answered in the manuscript. For example, why does the interference control not produce the same effect? That is, why is it supposed that rats associate stimuli A and B together within the same context but not another cues which are also subsequently presented in these contexts in the “practice phase”? Is it just because rats experience objects A and B first and 15 minutes closer together than the other objects which occur in a block?

Author Response 1.5. As noted in Author Response 1.2, subpoint 3, we do not, in fact, assume that A and B are associated to one another. In fact, the whole reason why we separated the presentation of A and B into different stages, separated by 20 minutes was to reduce the likelihood of such an association. This design choice was prompted by the human literature on retrieval-induced forgetting, which has found that associations between A and B reduce the forgetting effect we were predicting. The reviewer is inferring that our intention was to link A and B, based on viewing the paradigm through the lens of retrospective revaluation, which was not our intention.

What we do assume is that the Context gets associated to object A; and then later on, the same Context gets associated to object B, creating the retrieval competition situation illustrated in the diagrams from the paper. This is the situation that is analogous to the one studied in humans, which prompts retrieval-induced forgetting. The critical causative factor that induces forgetting is the attempt to selectively retrieve object A when presented with A during retrieval practice in the same context. We assume that the context cue elicits activation of A and B simultaneously, and that isolating A in memory leads to the inhibition of B, in service of overcoming retrieval competition. Note that in the interference control condition, because all objects are novel, there is no episodic retrieval demand in play---rather, the task simply involves encoding new objects. In

the human literature, it has been shown that retrieval practice—and not new encoding---induces forgetting. So, this finding is very much in line with the human literature.

Reviewer Comment 1.6. In the manuscript there is a suggestion that rats may experience object A or B first (i.e. p4, line 117)? Is the description of the protocol in the figures just an example of one set of counterbalancing? If so, can we see the data on conditions where B is given first, and then where A is given first?

Author Response 1.6. The reviewer is correct--the figure is only an example, and which object was selected for practice varied across animals. There were no differences in the effects depending on which object was selected ($p=0.8687$, $t=0.17$, unpaired t test; RP (object B first) vs RP (object B second); $n=6$).

Reviewer Comment 1.7. Unless I am missing something from the argument, it is not entirely clear to me why the interference control does not produce the same effect and specifically why objects A and B are associated together. Usually, rats would be exposed to stimuli together to produce these effects (i.e. object A and B in arena 1 at the same time).

Author Response 1.7. Viewed from the standpoint of retrospective revaluation, we can understand why the reviewer might be confused by this. However, this is not a retrospective revaluation study, and was never intended to be, as described in Author Response 1.2. Our rationale grew out of the human retrieval-induced forgetting literature, as described in Author Response 1.5 and in the paper itself.

Reviewer Comment 1.8. Would the authors predict that object A alone in arena 1 during the practice phase would be enough to induce the forgetting of B? Or does it need to be compared with another stimulus to be “retrieved”?

Author Response 1.8. Presenting object A alone in the arena during the retrieval practice phase may be sufficient to produce the effect—we don’t know. However, in designing our protocol, we sought to be very conservative and to ensure that the retrieval practice task required episodic retrieval from the rats, to match the human designs. To do this, we pitted A against a second object which had been pre-familiarised in a different context (see paper and methods). As such, each retrieval practice trial was composed of two familiar objects, only one of which was exposed in the current context. Thus, any preference for the contextually novel distractor would reflect a discrimination that was based on remembering that A was experienced in the current context (note: we are building on existing findings here which show that such “contextual novelty” is something that rats show and that is hippocampally dependent).

In a nutshell-we used the design we thought provided the strongest test; presenting a single object without a foil might well have worked but would not have been as likely to spontaneously entrain episodic retrieval.

Reviewer Comment 1.9. The experiment testing for the context generalizability of the forgetting effect is interesting (though it suffers from the issue I describe above also). Specifically, the authors find that the “adaptive forgetting” effect generalizes across contexts. This suggests that the rats really do “forget” the object. That is, these results cannot be described in terms of a simple reduction in context-object association because if this were the case then the effect would not occur in a new context. The critical control for this is the interference control, where rats just exposed to other objects in the same context which are not “related” to the critical forgotten object do not show forgetting of the critical object- as an aside I thought the authors might better

make this point about the importance of the interference control in this particular experiment. This is an important point because it tells us something about the associative structure of what is going on. I thought the authors might better focus on this point and discuss the consequences that this experiment has for our understanding of the associative structure of their effect (and generally retrospective revaluation).

Author Response 1.9. We agree that the cue-independence experiment does illustrate that the effect is not associative in nature. Indeed, the logic of this design was first introduced in Anderson & Spellman (1995)⁵ to rule out associative accounts of retrieval-induced forgetting, echoing the reviewer's reasoning. As a separate aside, however, the critical variable is not whether the objects occurring during practice are related to the inhibited item, but (a) whether they shared exposure in a common context, and (b) were given episodic retrieval practice in the interim.

Regarding the potential implications of this finding for RR, we hesitate to make strong claims because, as noted in our prior responses, it is not our intention to address RR in this paper, and we feel that the case for the analogy between the phenomena cannot be established clearly and persuasively in the context of the studies we have.

Author Action Taken 1.9. We took the reviewer's advice and tried to emphasise the importance of the cue-independence finding with respect to ruling out associative accounts. This change can be found in the final paragraph of the cue-independence section. First, we separated off the final sentences of the preceding paragraph to make this new paragraph, to highlight and emphasise the conclusion. Second, we explicitly state that the finding is theoretically important because it rules out purely associative accounts.

Reviewer Comment 1.10. Another novel point of the manuscript is that the role of medial prefrontal cortex in this effect is transient. That is, cFos indicators of neural activation in medial prefrontal cortex in the adaptive forgetting effect occurs in the first encounters with the object during the practice phase but indicators of activity taken later in the practice phase do not elicit significant results. There is probably a more elegant way to have demonstrated this (e.g. muscimol infusion after two practice sessions have taken place), but the point is novel and important.

Author Response 1.10. The reviewer is right that the muscimol manipulation would have been elegant...arguably more elegant than what we did. In future replications of this finding, we might pursue that idea. Our intention in this study, however, was to create a rodent analogue of a repeatedly observed finding in the human literature, that early retrieval practices engage PFC control mechanisms more than later ones, with the extent of the decline predicting successful forgetting of the competitor. c-Fos imaging was our closest approximation to replicating this finding.

Reviewer Comment 1.11. It is consistent with other work looking at medial prefrontal cortex and the resolution of response competition (in particular, prelimbic cortex on goal-directed responding, attentional tasks etc.), where the role of prelimbic cortex is often transient. This again tells us something about the nature of the "adaptive forgetting"/retrospective revaluation effect. Sometimes the role of the prelimbic cortex is transient and sometimes it is not. For example, the prelimbic cortex is involved in both the acquisition and expression of context-dependent associations. So, the transient nature of the medial prefrontal cortex in this effect suggests early encounters with the competitor object elicit the inhibition effect that wanes across time. The

authors again might spend more time talking about the significance of this point and what it tells us about the associative nature of the effect.

Author Response 1.11. We thank the reviewer for these comments. We agree the declining involvement of the mPFC over repetitions is important to the interpretation of the effect. Our interpretation is that the initial competition, triggered during early retrieval practice trials, triggers engagement of an inhibitory control mechanism mediated by mPFC, to inhibit the competing trace; the persisting effect of this inhibition reduces future competition, and any further need to engage the mPFC is consequently reduced. This interpretation is in line with the human literature on retrieval-induced forgetting, which has linked this decline to the extent of forgetting of the competitor on later tests. It is also in line with rodent work on the role of this structure in resolving retrieval interference, in addition to whatever role it may play in forming new associations (e.g. Peters et al.)⁶.

Author Action Taken 1.11. We see the reviewer's point that we did, in fact, understate the importance of this finding about a time-limited role of the mPFC, and we thank the reviewer for calling our attention to this. As a solution, we separated off the major theoretical point we wanted to make from what had been the final paragraph of the c-Fos section and made a new paragraph. In this new paragraph, we augmented our discussion by highlighting why the juxtaposition of the effects of the interference control and retrieval practice conditions was important, and what we think it means. We considered adding further text talking about other time-limited effects in the mPFC (e.g., the Tran-Tu-Yen et al. paper)⁷, but in the end, we elected not to because it would further lengthen the paper and deflect from our point (in part because of substantial differences in paradigms and time scales of the earlier transient effects, necessitating qualification and discussion). If the reviewer wishes, we will reconsider this decision and add that discussion, if they feel it is essential.

Reviewer Comment 1.12. The final significant point I would raise is that the manuscript is very hard going. Indeed, it took me significantly longer than usual to review it. I found that the graphical representations of the data were quite confusing and I kept needing to go back and read the results to understand exactly what the authors had found. For example, in Figure 3C there is suddenly a reference to non-competitor F which is not referenced in the figure illustrating the design.

Author Response 1.12. We are truly very sorry if we did anything to create confusion for the reviewer. The reviewer is right to point out that 7 experiments with multiple conditions and varying designs is inevitably going to require some effort from readers and that it is very important that we optimise the presentation to reduce reader burden. We worked quite hard on the initial submission to be as clear as we could, but clearly we had more work to do.

Author Action Taken 1.12. We have corrected the issue raised by the reviewer concerning Figure 3C and we have gone back and tried to improve our presentation and clarity as much as possible. We considered eliminating the graphical figures because the reviewer thought they were confusing, but one of the other reviewers said the opposite thing—that their understanding of the design hinged on the figures—and wanted more of them! In the face of this contradictory advice, we tried to compromise by (a) adding new graphical figures for all studies, but (b) putting the new ones in a supplement to not overburden readers. In addition, references to object F were added in the figure caption pointing at the supplementary material. We have also mention this object in the main text,

Reviewer Comment 1.13. Generally, I don't think the results are represented graphically in a straight-forward manner. Further, the wording describing the effects is very verbose and confusing. And the results section and description of the methods is entirely too long. I think the authors could significantly shorten these sections and focus on the main points of their data or else I am concerned that the main points of the data will not get across to their readers.

Author Action Taken 1.13. We have revised the methods section of the paper to reduce redundancy.

Reviewer Comment 1.14. In summary, I think there are some important points that come out of the manuscript. If the manuscript goes to publication here I think the authors need to recognize the extensive associative literature that has come before this (i.e. retrospective revaluation) and the associated controversy related to the associative nature of this effect and focus more on the unique points related to their data (i.e. context generalizability of their effect, and the transient role of the medial prefrontal cortex in this design).

Author Response 1.14. We thank the reviewer for the considerable effort that they put into reading and reviewing our manuscript, and for highlighting its potential connection to RR. We have now incorporated a paragraph on RR in the discussion to encourage readers to draw those links and possibly pursue them. Although this is not the full-scale re-contextualization of this work in terms of RR that the reviewer might have wanted, it does achieve the goal of calling attention to the relationship so others can pursue them if they wish. This seems like the right balance between preserving the true intentions of our work, and its genesis in the human literature, and honouring the reviewer's request.

Reviewer Comment 1.15. It is also very important that the authors make it clear why objects A and B by necessity become associated in a way that does not generalize to the other object in the same arena in the interference control condition?

Author Response 1.15. As noted in previous remarks, we do not assume that A & B become associated—quite the opposite.

Author Action Taken 1.15. To address this issue, we added a sentence at the end of the first paragraph in the results section, highlighting that we deliberately separated the presentation of objects A and B in an effort to avoid associations between them, following work on retrieval-induced forgetting that shows that such associations reduce the effect. Hopefully, this will signal to RR readers that we are not seeking to associate these objects and that this is not part of our thinking about what produces the phenomenon.

Reviewer Comment 1.16. Below is a (non-exhaustive) list of references which are related to the present manuscript and should be discussed and cited in the manuscript:

Dickinson, A. (1996). Within compound associations mediate the retrospective revaluation of causality judgements. *The Quarterly Journal of Experimental Psychology: Section B*, 49(1), 60-80.

Le Pelley, M. E., & McLaren, I. P. L. (2001). Retrospective revaluation in humans: Learning or memory?. *The Quarterly Journal of Experimental Psychology: Section B*, 54(4), 311-352.

Corlett, P. R., Aitken, M. R., Dickinson, A., Shanks, D. R., Honey, G. D., Honey, R. A., ... &

Fletcher, P. C. (2004). Prediction error during retrospective reevaluation of causal associations in humans: fMRI evidence in favor of an associative model of learning. *Neuron*, 44(5), 877-888.

Ghirlanda, S. (2005). Retrospective reevaluation as simple associative learning. *Journal of Experimental Psychology: Animal Behavior Processes*, 31(1), 107.

San-Galli, A., Marchand, A. R., Decorte, L., & Di Scala, G. (2011). Retrospective reevaluation and its neural circuit in rats. *Behavioural brain research*, 223(2), 262-270.

Tran-Tu-Yen, D. A., Marchand, A. R., Pape, J. R., Di Scala, G., & Coutureau, E. (2009). Transient role of the rat prelimbic cortex in goal-directed behaviour. *European Journal of Neuroscience*, 30(3), 464-471.

Sharpe, M. J., & Killcross, S. (2015). The prelimbic cortex uses higher-order cues to modulate both the acquisition and expression of conditioned fear. *Frontiers in systems neuroscience*, 8, 235.

Stout, S. C., & Miller, R. R. (2007). Sometimes-competing retrieval (SOCR): A formalization of the comparator hypothesis. *Psychological Review*, 114(3), 759.

Melchers, K., Lachnit, H., & Shanks, D. (2004). Within-compound associations in retrospective reevaluation and in direct learning: A challenge for comparator theory. *The Quarterly Journal of Experimental Psychology Section B*, 57(1), 25-53.

Author Response 1.16. Thank you for this helpful list!

Author Action Taken 1.16. In our new paragraph in the discussion, we cite several of these papers, and others that we found.

Reviewer 2

Reviewer Comment 2.1. The authors report findings from a rodent model of adaptive forgetting, the retrieval-induced forgetting effect. This is an excellent paper that convincingly demonstrates that the retrieval of competitor memories is inhibited by retrieval practice of other memories from the same context, a phenomenon which has been extensively studied in human subjects but has received very little attention in animal models. Indeed, the notion that retrieval inhibition plays a key role in forgetting has not really penetrated the animal behavioral neuroscience of memory community, much to the detriment of the study of the neuroscience of memory in my opinion, so this paper has high potential impact.

Author Response 2.1. We thank the reviewer for their nice remarks about the paper! We are very happy that they think that the work has potential for high impact on the field.

Reviewer Comment 2.2. The experimental design is very clever, taking advantage of spontaneous object investigation and the fact that rodents readily associate objects with the context where they were encountered. Importantly, the authors have carried out the appropriate controls to clearly rule out alternative explanations (i.e. other than retrieval-induced inhibition of the competitor memory), including time and interference controls, but also demonstration of cue independence, specificity to competitor memories and durability of the effect out to 24 hours. These controls are crucial: they not only rule out trivial (but unlikely) alternatives, but they really define the nature of the memory effect and distinguish it from a variety of other plausible memory phenomena. The inclusion of these controls is not surprising given that the senior author is the

leading authority on this phenomenon as it has been studied in humans. Nevertheless, it is notable that this particular experimental design allowed the authors to include the relevant controls in a manner that kept other aspects of the task comparable to the conditions that produced the basic retrieval-induced forgetting effect.

Author Response 2.2. Thanks very much for highlighting these features of the work—we agree that the control experiments are crucial, not only in ruling out less interesting explanations, but also in making the scientific case for functional parallels with the human phenomenon. That, in fact, is a major aim of the paper—taking attributes from the human literature and testing them in rodents.

Reviewer Comment 2.3. Finally, these experimental procedures provide an excellent approach for studying these adaptive forgetting/retrieval inhibition effects using a variety of invasive manipulations that cannot be done with human subjects. The authors have already performed two of these, muscimol inactivation and measurement of IEG expression in the medial prefrontal cortex, and it is likely this paper will spur additional research.

I am generally enthusiastic about the paper, but I do have a few concerns which should be addressed.

Author Response 2.3. The reviewer here correctly identifies the ultimate goal of the work—to enable researchers to use the impressive array of invasive techniques only available in animals to study what we view as a crucial model of forgetting that has direct relevance to humans. We are thrilled at the potential of this work to inspire a surge of research in neurobiology of forgetting—an understudied area—and to make significant advances in this fundamental phenomenon. We are delighted that the reviewer sees this potential!

Reviewer Comment 2.4. A similar phenomenon was reported in rats by another laboratory (Wu et al, 2014, Hippocampus) and although the authors cited the paper, they should discuss it in more detail. The previous paper reported that retrieval inhibition occurred in an olfactory learning task and that medial prefrontal inactivation (and hippocampal inactivation) disrupted this effect. However, in my opinion the previous finding should not detract from the novelty of the present results. The Wu et al paper had a number of inadequacies, including that it used an instrumentally reinforced behavior (rather than a spontaneous behavior as in the present study), which could have unknown effects, and it was impossible to rule out alternative explanations for the finding. It did not include the critical controls that really define the phenomenon, as described above. For these reasons, I think the present study is not simply an extension of previous work, but a novel and much more convincing demonstration of retrieval inhibition in rodents. Nevertheless, the previous paper is, at minimum, consistent with the present results and the broader literature on retrieval inhibition. The authors should more explicitly discuss the previous findings and theoretical framework for mPFC inhibitory control, and how they relate to the present results.

Author Response 2.4. We agree with the reviewer fully that the Wu et al. paper would be an excellent study to discuss and are very happy to do so. We agree that although the earlier study lacked many of the key controls present here, it nonetheless provides converging evidence from a different method.

Author Action Taken 2.4. To address the reviewer's request, we have now inserted a new paragraph in the Discussion of the paper, highlighting the Wu et al. work, and discussing it in relation to the current findings. (see paragraph 4, discussion).

Reviewer Comment 2.5.

Minor Concerns

The authors say on p 4, line 124 "... they had retrieved an episodic-like, place-specific memory ...". Why 'episodic-like'? This task doesn't seem to meet the typical 'what, where, when' requirements that other authors have suggested for 'episodic-like' memory (a phrase I find a bit too equivocal anyway, but others disagree). Also, 'place-specific' may be confusing if the authors actually mean context-specific, since rodents can learn to match objects to contexts and to specific locations within a context (i.e. 'places').

Author Response 2.5. We believe that the retrieval process engaged during the retrieval practice phase approximates what, in the human literature, is known as episodic recollection—the recall of a specific event that took place in a certain location at a certain time. Because rats discriminate between two equally familiar objects on the basis of where they saw them (preferring the one not seen in the current context), context-based retrieval is very likely behind this tendency, consistent with the recollection view. This was the reasoning behind our use of "episodic-like" memory as a term. However, we take the reviewers point. Although our task arguably taps the "what" (i.e., the object), and "where" (i.e. in this context) requirements, it does not clearly isolate the "when" feature. So, we only have partial support for using this term. We also agree with the reviewer's comments about the potential confusion created by "place specific"

Author Action Taken 2.5. We eliminated the "episodic like" and "location specific" terms from this line and rephrased as "context specific". We also scanned the rest of the manuscript to make sure we use this term consistently.

Reviewer Comment 2.6. The authors sometimes use the term 'adaptive forgetting', sometimes 'inhibition', and sometimes 'retrieval-induced forgetting'. I have no objection to any of these, but I think it would be helpful to be clear and explicit about what each of those terms means and whether they're being used interchangeably, especially for readers who are not familiar with this literature.

Author Response 2.6. We thank the reviewer for this helpful comment. Indeed, we have not been sufficiently attentive to clearly defining and distinguishing these terms. Doing so will be very helpful to readers.

Author Action Taken 2.6. We have now revised the paper in which we tried to adopt the following conventions in the use of terminology. First "adaptive forgetting" is a potentially general category of forgetting processes that are helpful to organisms' adaptation; second, "retrieval-induced forgetting" is a particular empirical phenomenon which may be a useful model of adaptive forgetting, and third, "inhibition" is the mechanism—inhibitory control—that underlies retrieval-induced forgetting. In revising the paper to adopt this scheme, we now see that we had not consistently applied this, nor had we clearly signalled the "superordinate/subordinate" relationship between adaptive forgetting and RIF. Now, we state this explicitly, in the introduction, that RIF may be a useful model of adaptive forgetting, which signals that it is one form of adaptive forgetting. We thank the reviewer for calling our attention to this.

Reviewer Comment 2.7. mPFC lesions can cause hyperactivity and perseverative responding. Can the authors rule out the possibility that these non-mnemonic effects of the muscimol influenced their results?

Author Response 2.7. Our data rule out influences of hyperactivity or perseverative responding on encoding or on the final memory test in which we measured retrieval-induced forgetting. Indeed, the potential influences of the drug on performance is why we inserted a 24-hour delay between the retrieval practice phase and the final test phase. Because we administered muscimol after encoding, but before retrieval practice, and because 24 hours is sufficient for muscimol to “wash out”, we are confident that neither encoding nor the final test is not contaminated by these issues. Consistent with this, the only condition affected by muscimol is the competitor condition—neither the interference control nor the time control conditions showed an effect of this drug, suggesting that behaviour at test was largely similar to when vehicle was administered.

We also examined evidence for non-mnemonic effects during the retrieval practice phase, by scrutinizing total exploration time during this phase, and rats’ novelty preference behaviour. As reported in the tables provided in the supplementary materials, although there was a numerical tendency for less overall exploration in the muscimol condition than in the saline condition during retrieval practice, this difference was very far from significant. Perhaps more importantly, during retrieval practice, rats significantly discriminated familiar from contextually novel objects in both the muscimol and saline conditions, indicating that they both performed the task as we had hoped.

Author Action Taken 2.7. Although the data do not support the possibility raised by the reviewer, we agree that it is worth including reference to such possibilities in the text, when discussing Experiment 6. This will make readers aware of this as a potential interpretive issue (even if the data don’t support it) and allow readers to judge for themselves whether it matters to the interpretation of the effect. It may prompt readers who follow up on the work to specifically investigate this issue, which would be a good thing to do. Thanks for the suggestion. The relevant edits that highlight the hyperactivity issue can be found in paragraph 2 of the section reporting Experiment 6.

Reviewer Comment 2.8. The cFos data are interesting and the differences between early and later practice fit very nicely with the author’s theoretical framework on the role of the PFC. However, although the peak cFos expression may occur at 90 min, I assume there is a gradient of expression surrounding that time point and I think most. It looks like all of the rats continued training and testing beyond the time point of interest for cFos expression. I would like to see a discussion of how these subsequent events might have contributed to their cFos measures.

Author Response 2.8. For the cFos experiment, the behavioral protocol was changed in order to minimize any contribution to our cFos measurements that was not specifically related to the early or late practice sessions. For example, the encoding phase took place the day before the practice phase, so any cFos activation related to encoding would have already returned to basal levels at the beginning of the practice phase. For the "Early" condition, the rats were sacrificed 90 min after the second practice trial, immediately after the third practice trial and the test. Increases in cFos mRNA have been observed around 20-30 min after different types of manipulations⁸⁻¹⁰ and increases in cFos protein have been detected between 60 and 90 min post-induction^{8, 11-14}. Not many studies did a full time course of cFos increase and decay, but a few have found that cFos mRNA and protein returned to basal levels around 120 min after the behavioral manipulation or electrical stimulation^{9, 10, 12, 13, 15}.

Given this evidence, it is highly unlikely that cFos protein from the third practice trial or the test contributed to the cell counts in the "Early" condition. In the "Late" condition, animals were sacrificed around 90 min after the third practice trial, immediately after the test. The peak of cFos counts came from the putative activity that the mPFC had during the third trial and any possible

residual cFos expression coming from the first and second practice trials. However, it is likely that the cFos coming from these early trials had little contribution, since, in this condition, we measured cFos around 180 min after the second practice trial and around 195 min after the first one, when cFos coming from neuronal activity would have decreased^{9, 10, 12, 13, 15}. In fact, cFos counts in the "Late" condition are indistinguishable from the ones in the TC condition when the practice phase was absent. This suggests that cFos is significantly high 90 min post-practice in the "Early" condition, like after many other behavioural manipulations, but then returns to basal levels 180 min post-practice and does not affect cell counts in the "Late" condition. It is worth mentioning that even if cFos cell counts in the "Late" were accumulating some staining signal coming from practice trials 1 and 2, there are still significant differences between the "Early" and "Late" conditions for the RP animals.

Author Action Taken 2.8. We agree with the reviewer that it is worthwhile to discuss the above considerations, so readers can have a better understanding of the reasoning behind our procedural control. We therefore have discussed the possible time course of cFos expression and how our design largely rules out any cumulative effects in our measurements in the third paragraph under the heading "Adaptive benefits of forgetting".

Reviewer Comment 2.9. The authors might consider putting task schematic illustrations in figures 4 and 5, like those in earlier figures. This would be particularly useful for the cFos study.

Author Response 2.9. We had actually included these figures in our initial drafts of the manuscript. In the end, we took them out because we felt that it made the manuscript "figure heavy".

Author Action Taken 2.9. We agree, however, that having the figures someplace might benefit some readers. To address the reviewer's point, while keeping the manuscript streamlined, we have included all of the procedural figures for every experiment in the supplementary materials and refer to this now in text. This solution was also inspired by the request from Reviewer 3 that we duplicate the figures in the supplement to make it easy for readers of the detailed methods to understand the procedure.

Reviewer Comment 2.10. If practical, the authors might consider putting additional descriptive labels on bar graphs (e.g. B vs C and J vs D under the red bars in Fig 2C, etc.) in addition to the current text.

Author Response 2.10. Thanks for the suggestion....that's easy to do.

Author Action Taken 2.10. Because Fig 2C has two panels, we put B vs C under the left panel, J vs D under the right panel.

Reviewer 3

Reviewer Comment 3.1. In a series of experiments using the novel-object paradigm in rats, Bekinschtein and colleagues sought to test the hypothesis that repeated exposure to a familiar object in a familiar context will lead to retrieval-induced forgetting of a previously learned object in that same context due to a competition process that is mediated by the medial-prefrontal cortex (mPFC). They propose that retrieval-induced forgetting is the result of active inhibitory control.

Their retrieval-induced forgetting hypothesis was supported in an elegantly designed Experiment 1, and controlled for in a series of other experiments, though the claims of an active inhibitory control mechanism, paralleling that found in humans, remain somewhat speculative.

Author Response 3.10. We thank the reviewer for their nice remarks about the work.

Reviewer Comment 3.2. While I found the main results of the paper quite interesting, I admittedly found it a bit of a struggle to read. As a general comment, though the methods were described in detail on pg 17, the multi-stage conditions for 7 experiments, each with multiple objects and contexts made it very difficult to follow the procedures. Having read the detailed methods section on pg. 17 prior to the results, I was working very hard to keep all the experiments, contexts, phases, trials, and timelines straight. The schematics of the experimental designs referred to in the results (Fig. 1A, 2A, 3A) help to make sense of it all. For readers who review the specific methods before the results, it will be of great use to refer to these study design schematics within the detailed methods section.

Author Response 3.10. That's a great suggestion. We hadn't anticipated that readers might start with the detailed methods before reading the paper, but in retrospect, this is an obvious possibility. The figures really do help. To address this point, we actually thought it was a good idea to put the figures in the methods too, so that it is self-contained and readers don't have to flip back to the paper.

Author Action Taken 3.10. We have added to the supplementary methods the schematics of the experimental designs for each experiment—including schematics for the later experiments, which we hadn't included in the original paper. This should provide a great deal of reader support, hopefully.

Reviewer Comment 3.3. My primary reservations with the paper relate to the claims that the retrieval-induced forgetting is (1) an active inhibitory control mechanism mediated by the mPFC, and (2) that this is a species-general phenomenon.

Based on the methods used (observation of exploratory behavior in a novel-object paradigm, silencing of mPFC, and IEG expression of c-Fos in mPFC following retrieval), it is not clear how the findings represent 'active inhibitory control' over forgetting. The authors have nice, well controlled findings in Experiment 1 in which repeated exposure to Object-A following exposure to both A & B within a context results in less exploration of familiar/distractor Object-B at a later timepoint when tested in that same encoding/sampling context. Though the results of Experiment 6 (mPFC inhibition) and Experiment 7 (mPFC IEG expression) suggest that the mPFC plays a role in mediating this retrieval-induced forgetting, it remains a stretch to state that inhibitory control is the specific mechanism, as this was not explicitly tested. To support this claim, the authors should provide a direct measure of inhibitory control mechanisms in the present study.

Author Response 3.3. We understand the reviewer's point of view about whether our data supports inhibitory control and trying to understand this viewpoint has been helpful. Part of the problem, we think, is that we are using quite different standards with which to evaluate the involvement of inhibition, and we didn't explain our own standards clearly enough. We apologise for this, and clarify the basis of this conclusion here, for the reviewer's consideration. But before elaborating on how we view our evidence for inhibitory control, it's important to address what may be a miscommunication about our intended mechanism, which we describe next.

3.3.1. Possible miscommunication about the intended mechanism. After reading the full range of comments by Reviewer 3, we discerned what may be an assumption about our mechanism that we did not intend, owing to lack of clarity in the way that we presented our argument. Based on several comments, especially Reviewer comments 3.4 and 3.5 below, it appears that the reviewer assumes that we are claiming that the prefrontal cortex continues to exert a sustained role in producing forgetting on the final memory test in which RIF is measured. That is, the reviewer may have interpreted us as claiming that (a) retrieval practice trains a forgetting process that involves an interaction between prefrontal cortex and competing memories, and (b) to observe forgetting on the final test, that prefrontal cortical mechanism must continue to be engaged in suppression or alternatively must instead must be re-engaged at test to create the forgetting. For example, the reviewer might suspect that we are proposing the formation of an inhibitory association during retrieval practice that is re-engaged by the cue at test, analogous to what data suggest seems true of extinction. This interpretation of our proposal is entirely reasonable and is not really ruled out by anything we had said in our paper, and has ample precedent, so we can understand how the reviewer might have come to that view.

To be clear--this is not our intended mechanism. Rather, our account states that (a) during retrieval practice, competitors are suppressed by inhibitory control mechanisms, (b) this momentary application of control triggers a change in state in the competitor that renders it less accessible in an enduring way, without the need for ongoing control or prefrontal involvement, and (c) this change in state can be measured on subsequent retention tests. By this account, one could lesion the mPFC just before the final test and the forgetting would still occur because the disruption has been done by its recruitment during retrieval practice. Thus, the aftereffects of inhibitory control at retrieval practice have an enduring effect on episodic memories. This view has critical differences from the one in the preceding paragraph that would have significant bearing on whether our experiments provide sufficient information to evaluate the role of inhibition. Bearing this conceptualization in mind, the following evidence supports these ideas.

3.3.2. The Nature of the Evidence We Have. It is possible that the foregoing assumption may have been behind some of the reviewer's hesitancy (if not, we apologise for our misunderstanding). Another possibility is that Reviewer 3 may have been focusing on the need for different types of evidence (e.g. proof that neural inhibition is involved, or a direct connection to molecular forgetting mechanisms) than we intended to provide. Our intention was to provide a clear set of cognitive/behavioural tests to evaluate predictions of the inhibitory control hypothesis of retrieval-induced forgetting in rodents. To do this, the current studies built on methods and logic that have been developed by the senior author, who first reported retrieval-induced forgetting in humans in 1994, and who developed the methods and arguments now routinely used in cognitive psychology. From the perspective of that cognitive literature and its established methods on this subject, the present study addressed the role of inhibitory control in the phenomenon in very specific ways (as recognized by Reviewer 2) that, from a cognitive standpoint, test the role of inhibition very directly.

Our studies tested distinctive behavioural predictions that were derived based on alternative inhibitory and non-inhibitory theories of retrieval-induced forgetting and tested thoroughly over a decade (See Anderson & Bjork, 1994; Anderson & Spellman, 1995; and Anderson, 2004, for extensive overviews)^{2, 5, 16}. The theoretical proposal, in the cognitive realm states that (a) when a particular memory is sought, the cue or cues may elicit not only the target memory, but also many competing traces, (b) to resolve that competition, an inhibitory control process suppresses competing traces to facilitate selection of the desired target memory, and (c) the persisting after-effects of this suppression on competing traces make it harder to recall or even recognize the competitors on later tests. Note that we do not assume that the "persisting aftereffects" are

implemented by a sustained suppression; rather, the momentary suppression of a competitor is thought to have lasting effects on the integrity of the inhibited trace. The principal goal in this literature has been to differentiate that account from non-inhibitory mechanisms such as associative blocking, unlearning, context shifts, etc. By developing ways to evaluate these distinctive predictions in rodents, we were in fact providing a very strong test of the inhibitory control hypothesis of retrieval-induced forgetting.

The methods we used here do achieve those goals as well as they can be achieved at the behavioural/cognitive level. In these studies, we provide evidence for cue-independence, retrieval-specificity, interference dependence, and strength-independence, for example. The properties rule out a variety of non-inhibitory alternative accounts in terms of associative interference, unlearning, context effects, etc (see Anderson & Bjork, 1994, and also Anderson, 2004, for comprehensive reviews of theoretical accounts)^{2, 16}, which fail to make correct predictions about when forgetting will be observed in these studies. Moreover, we show that the amount of forgetting (a) is clearly more than what happens with the mere passage of time (so not just decay), (b) is clearly more than what happens with significant interference (interference control), (c) is clearly associated with active retrieval, and (d) reliant on a prefrontal structure that is implicated in inhibitory control over action (in the rodent literature), exactly as would be expected, and paralleling similar demonstrations in humans. We even can show that the involvement of the putative prefrontal control structure declines over repetitions, as it does in humans (Kuhl et al., 2007, *Nature Neuroscience*)¹⁷. These parallels are strong and nearly perfectly align this phenomenon to a large human literature pointing to a prefrontally-mediated, attention-dependent inhibitory control processes that disrupts competing memory traces in a cue-independent fashion.

3.3.3. *The Evidence that we Don't Have.* Comprehensive proof of inhibitory control in modulating forgetting would include other forms of evidence, including demonstrations of the pathways and neural dynamics by which the PFC modulates the accessibility of the competing memory, including proof that it involves neural inhibition that affects competitors. We present no evidence on these points. This was not our purpose in this paper. Our purpose was to take a functional phenomenon in humans and establish—with the types of functional level evidence used in humans—that the effect arises in animals, and obeys properties taken by many as distinctive cognitive evidence for inhibitory control. This effort was successful and sets the stage for the more refined evidence regarding the above points enabling the necessary work to be conducted. Thus, our view is that the present studies do test this mechanism directly, at least in terms of widely discussed behavioural criteria.

Author Action Taken 3.3. Because our argument was not sufficiently clear and because of the potential for misunderstanding about the longevity of the effect, and need for PFC at test, we have undertaken numerous revisions to the manuscript to clarify and emphasise these points. Doing so has revealed to us just how unclear our initial draft was, for which we apologise.

Specifically, we have now stated very clearly that we believe that (a) mPFC is acting as an inhibitory control mechanism, (b) mPFC interacts with other sites such as the hippocampus to achieve inhibition, and (c) that the initial application of inhibition triggers a forgetting process which persists over time, without any needed involvement of the mPFC. Hopefully, these changes will clarify our theoretical view, and prevent future readers from misunderstanding the proposal. These changes can be found (1) at the end of the second paragraph in the Results section, (b) at the end of the single-paragraph section entitled “Durability of the forgetting”, (3) end of paragraph 1 in section entitled “Role of the prefrontal cortex in driving adaptive forgetting” (4) paragraph 2 of section entitled “Adaptive benefits of forgetting” (hippocampus now mentioned)

and (4) towards the end of paragraph 2 in the Discussion, where again, fronto-hippocampal interactions are mentioned.

In addition, anticipating the possibility that some readers may prioritise other forms of evidence for inhibitory control, we have emphasised that we are seeking high level functional/cognitive parallels that are indicative of inhibitory control. This can be found in the final paragraph of the introduction. In addition, we have softened our conclusions about inhibitory control, summarising by saying that our findings are *consistent with a role of inhibitory control in forgetting*, rather than stating that this has been proven. This can be found, e.g., in the third sentence of paragraph 2 in the Discussion, and in multiple edits throughout the paper.

Reviewer Comment 3.4. One of the main objectives of the study was to determine if the prefrontal cortex plays a conserved role across species in mediating active forgetting. In the mPFC/muscimol experiment (Experiment 6), it is specified that rats were assigned to one condition (RP, IC, or TC) for each a saline and a muscimol test. This experiment did support the proposal that the retrieval-induced forgetting is mediated by mPFC, as silencing this region reduced forgetting of the distractor object during the test session. It was less clear what was gained by Experiment 7, in which expression of c-Fos was observed following retrieval of the highly familiar object, but, from what I could tell, mPFC c-Fos expression was not tested following the actual test phase, which would allow you to examine the activity of the mPFC at the time during which the retrieval-induced forgetting is observable.

Author Response 3.4. This reviewer comment was helpful in clarifying why the reviewer might be hesitant about our conclusions about inhibitory control. In the final sentence, the reviewer flags up, as a problem, the fact that we did not measure c-Fos after the final test, to measure mPFC activity that would have taken place during the test itself. At the same time, the reviewer doesn't see the diagnostic value of measuring this effect during the retrieval practice phase itself, and how the fact that the effect declines adds value to the argument. Taken together, these comments suggest that the reviewer interprets us as proposing that the mPFC remains involved in inhibiting memories on the final test, and that that is where the critical "action" is. Given that perspective, we can fully understand the reviewer's hesitancy in accepting our conclusions, given that we never actually measured c-Fos during the final test.

As noted in reply 3.3.1., it was not our intention to argue that inhibition is being produced on the final test, by learned inhibitory associations. Rather, our proposal is that (a) mPFC is engaged, as a control process to inhibit competitors during retrieval practice, (b) successful inhibition of competitors disrupt them, reducing their accessibility chronically, without any further need for engagement by the PFC, and (c) given that, on retrieval practice trial #1 for a given item, that the competitor is inhibited, the mPFC control process is no longer needed on later trials, because of the persisting effect, which reduces competitiveness. Thus, mPFC involvement should (a) decline over retrieval practice trials, and (b) be unnecessary on the final test, given the persisting disruption. According to this mechanism then, Experiment 7 confirms the predicted decline in c-Fos over retrieval practice repetitions (which conceptually replicates human imaging work, which links this decline to subsequent forgetting).

Author Action Taken 3.4. As noted in Author Action Taken 3.3., we have revised the manuscript to ensure that other readers don't also come away with the impression that the reviewer may have about our proposed mechanism.

Reviewer Comment 3.5. Instead, the results of Experiment 7 show a decline in mPFC expression of c-Fos over repeated retrieval practice sessions of target Object-A, suggesting, in line with what

is found in humans, that the mPFC becomes less active as an item becomes more familiar. The authors should provide some supporting evidence from other studies in rodents as well that support the weakened dependence on mPFC for familiar items. To support the argument that the mPFC is supporting retrieval-induced forgetting of the distractor item following repeated exposure to the target item, an extra c-Fos experiment should be added, where c-Fos expression in mPFC is analyzed following retrieval of distractor Object-B in the test phase (as well as a comparison of c-Fos expression following retrieval of target object-A in the test phase). This will provide a more direct measure of mPFC activity accompanying forgetting of the distractor item. Also, the full statistics need to be reported for the ANOVA results here, including the main effects, degrees of freedom, error (reported for second, but not first F value), and effect sizes.

Author Response 3.5.

3.5.1. The preceding comment helped us to understand where the reviewer was coming from, which, in retrospect, makes perfect sense to us now, and we see why the reviewer might have had that view. Specifically, the comment “To support the argument that the mPFC is supporting retrieval-induced forgetting of the distractor item following repeated exposure...” By this comment, it is clear that the reviewer is interpreting our mechanism as one in which the critical inhibitory action is happening at test, during which the medial prefrontal cortex is invoked to induce forgetting. Based on this interpretation, the reviewer’s hesitancy is perfectly reasonable.

As mentioned in the previous Author Responses however, we are making a different argument. In fact, we would not propose any enduring role of mPFC in the maintenance or expression of the RIF effect, as the reviewer supposes. So, the additional c-Fos study is not needed to test our hypothesis, though we think it would be an interesting study to conduct.

3.5.2. The Reviewer suggests that we provide supporting evidence from other studies in rodents that indicate a weakened dependence on mPFC for familiar items. However, we do not interpret the decline in mPFC involvement over repeated retrieval practices as mere familiarity of the repeated item. Rather, we interpret it to reflect the declining competition (over repeated practice trials) from competing items, owing to their successful and persisting suppression. To draw a comparison, in the human imaging literature, the extent in the decline in BOLD signal in PFC over repeated retrieval practices is very strongly associated to the amount of retrieval-induced forgetting on competing items on later tests (across multiple papers; e.g., Kuhl et al., 2007, *Nature Neuroscience*; Wimber et al., 2015, *Nature Neuroscience*)^{17, 18}, and is not associated to the facilitation or strengthening of the practiced item itself. These data suggest that it is not the familiarity of the practiced item per se that leads to reduced mPFC activity, but the diminishing need to engage the control process putatively mediated by the structure, due to successful forgetting. Indeed, it was this connection between the decline in prefrontal involvement and later forgetting that led Kuhl et al. to entitle their paper “Decreased demands on cognitive control reveal the neural processing benefits of forgetting.”

Given these considerations, we are not equating reduced mPFC activity with increased familiarity of the practiced items. For these reasons, it would not provide diagnostic support our argument to seek other rodent evidence that strived to show its declining involvement with familiarity. However, as far as we know, there has been no preceding effort to test the effects of repeated retrieval practice of objects on mPFC—it is not a typical procedure.

Author Action Taken 3.5. The reviewer’s description of declining activation across retrieval practice trials as reflecting increasing stimulus familiarity highlighted for us that we needed to address this reasonable possibility in the text. In the final paragraph of the Results, just before the

discussion, we included a paragraph pointing out that (a) there was little evidence of mPFC engagement during our interference control condition, on any trial, despite an equivalent degree of contextual novelty as would have been present during the first retrieval practice trial; as such, the interference control condition excludes the possibility that the mPFC is simply engaged by contextually novel objects or by encoding object-context associations. This pattern suggests that the mPFC decline in the retrieval practice condition is unlikely to simply reflect increasing familiarity of the repeated object. Rather, the mPFC is selectively engaged the need to resolve competition on the first practice trial. This is compatible with the view that this region is playing an active control role, as hypothesized.

Reviewer Comment 3.6. Additionally, in Experiment 7, the authors assessed c-Fos expression in mPFC following 2 (early) or 3 (late) retrieval sessions of the familiar item (Object-A), and found reduced c-Fos expression after late, relative to early, retrieval. Was the same retrieval timeline as described in Experiment 1 followed in Experiment 7 - specifically, was the 3rd retrieval practice trial (late) conducted only 15 min after the 2nd retrieval practice trial (early)? If so, would you not expect that the c-Fos expression from retrieval trial 2 would have been detectable, albeit at a lower expression level, 90 min after the 3rd trial, and this cumulative c-Fos protein expression across multiple retrieval events would limit your ability to determine the unique c-Fos expression induced by retrieval trial 3 vs. retrieval trials 2&3 in these animals? Given that the c-Fos levels were reported to be lower following the 3rd trial, these results would argue against this cumulative expression argument, but this potential confound should be addressed.

Author Response 3.6. Thanks for raising this point, which was also raised by Reviewer 2. This recurring point makes us realise that we have not been sufficiently clear, in text, about the controls in this study. In fact, the retrieval timeline in this Experiment was different from the one used in Experiment 1 to specifically avoid the problems mentioned by the reviewer. For Experiment 7, the behavioural protocol was changed in order to minimize any contribution to our cFos measurements that was not specifically related to the early or late practice sessions.

Author Action Taken 3.6. We have included in the supplementary methods section the schematics of the experimental design for Experiment 7 and all the rest of the experiments to help the readers follow the particular aspects of each design. See also response Author Response 2.8., where we address Reviewer 2's similar concern in detail.

Reviewer Comment 3.7. My second reservation with the study involves the heavy focus on drawing parallels between the findings in the present study, with analogous processes occurring in humans. Throughout the intro and results, the authors repeatedly state that their findings parallel retrieval-induced forgetting in humans. Similarly, the discussion emphasizes the parallels between the results of the present study and parallel findings in humans, suggesting that their finding of retrieval-induced forgetting is a 'species-general phenomenon'. While it is certainly valuable to make these connections across species, the authors refer largely to human literature in their discussion of their rodent results, with comparably little discussion of related findings within the rodent literature. The results of the present experiments need to be better supported by what is currently known (and unknown) about this process in the rodent brain before an extension to its applicability to humans and other species can be made.

Author Response 3.7. We strongly agree with the reviewer that it is, in general, wise to be cautious in generalizing phenomena across species. Indeed, it is precisely because we share the reviewer's values of conservatism that we performed 7 experiments rather than 1, before concluding that we were likely to be studying the same thing across species. We wanted to be quite confident before offering this to the field to consider.

Several points are worth making about our efforts here, and the context in which they were performed. First, there really is no existing literature on retrieval-induced memory impairments to refer to (as the reviewer asked us to do)—we are seeking to forge a new path here. An exception to this is a paper by David Smith and colleagues¹⁹, which we briefly mentioned in our original submission, but which we will now discuss in more detail in the current submission (see paragraph 4 of the Discussion).

Second, our approach to his line of work was entirely motivated by the human literature. We did not discover this phenomenon in rodents first, and then seek to retroactively relate it to the human literature. Rather, we set out, from the very first study, to test the existence of a species general retrieval-induced forgetting phenomenon. Every design choice in every experiment was a deliberate attempt to model the designs in human studies; the nature of the logic and questions in each study is modelled after the logic of human studies. We did not stop at the mere demonstration of retrieval-induced forgetting in experiment 1, but sought to demonstrate retrieval-specificity, interference dependence, and cue-independence, as found in humans; and our final lesion and c-Fos studies were motivated by analogy to human findings using imaging and TMS.

Third, the human phenomenon we sought to study in rodents was first reported by the senior author, who devised the logic and procedures that have been used in the human studies since 1994. This positions us very well to make careful decisions about design and interpretation that are maximally informed by the relevant human literature, as this author knows the many hundreds of papers in that domain very well.

Thus, although we agree quite strongly with the reviewer's stance on generalization, we have deliberately built our case in a way that was designed—from the very start—to address such concerns in the most rigorous way possible. Naturally, the reviewer and our readers are free to disagree with our conclusions, but they are offered in an honest and carefully considered way. We therefore ask the reviewer to allow us the latitude to make the case as we intended it to be made. There is real value in doing this-- the potential to advance a new set of research ideas into the field, and to link preclinical animal models to human experience.

Author Action Taken 3.7. We took another look throughout the paper to ensure that we consistently signal the appropriate concern for generalization. Although we did signal this concern often, we felt the paper would benefit from the addition of some sentences in the discussion highlighting that this remains a hypothesis, and also underscoring the need to explore the generalization. This can be found in the final two sentences of paragraph 2 in the Discussion. Moreover, we have now also included additional discussion of the David Smith's rodent work related to our point. This can be found in paragraph 4 of the Discussion.

Reviewer Comment 3.8. Finally, related to the active forgetting hypothesis across species, in the introduction, the authors do describe the mechanisms of active forgetting in drosophila, involving endocytosis of GluA2/AMPA receptors, which is a cellular mechanism that may very well underlie the forgetting seen in the present study, and which would be possible to test (though I recognize that was not the goal of the present study). The point here being, that they have proposed a testable mechanism in one species yet claimed that their results are parallel with a different mechanism in another species (active inhibitory control in humans). Given that these cellular mechanisms were not tested in the present study, it would be better to save the discussion of these potential cellular mechanisms for the discussion section, and to streamline the introduction and results to focus specifically on the results of their study (there are many

interesting ones), and to discuss the results of their experiments in the context of other relevant findings in rodents.

Author Response 3.8. The approach that the reviewer describes—of limiting the coverage of the cellular mechanisms in the introduction and elaborating in the discussion—was our intention, precisely because our data do not address the cellular mechanisms. We kept mention of this work to a minimum and introduced it only to put our work into the broader context of trends on research on active forgetting in the field. We were keen to draw in readers from multiple areas—from psychology, from systems neuroscience, and from cellular/molecular neuroscience—because we believe that that the current work provides a vehicle for vertical integration of work in these fields.

Author Action Taken 3.8. We retained the brief mention of cellular mechanisms in the introduction for the reasons mentioned above, though we tried to make the mention even briefer. We also recognize that by including mention of this work, we may inadvertently imply to readers that we will be reporting that kind of work. We therefore revised the introduction to more clearly signal the level at which we will be addressing adaptive forgetting.

Reviewer Comment 3.9. While, in the discussion, the authors do propose potential hippocampus-mediated inhibitory control cellular mechanism involving RAC-1 activity, they do not discuss potential cellular mechanisms within the mPFC, which is the focus of their studies. This should be addressed.

Author Response 3.9. The reviewer is correct that we do not discuss potential cellular mechanisms within the mPFC. We understand why the reviewer might suggest that the mPFC is the main focus of the paper, given that our main neural manipulations and measures targeted this structure. Moreover, based on (our guess) the reviewer's interpretation of our mechanism as involving mPFC inhibition during the final test (perhaps via learned inhibitory associations during retrieval practice), it makes further sense that this would lead them to expect a discussion of cellular mechanisms in mPFC, rather than the hippocampus—particularly since extinction studies focus on inhibitory learning in this region.

However, as mentioned in Author Response 3.4 and 3.5.1., our proposal is that the mPFC initiates a control signal that is temporarily engaged during retrieval practice to overcome competition between representations housed elsewhere (e.g. dorsal hippocampus), with the effects of this inhibition disrupting representations at those distal sites in an enduring way. As such, we believe that the site of inhibitory effects is likely to be dorsal hippocampus, not mPFC, which explains our focus on this in the discussion. Because our discussion of cellular mechanisms is focused on the mechanism by which plasticity might be affected (the targets of inhibition in dorsal hippocampus), and not on the cellular basis of the control process (the origins of control in mPFC), our discussion emphasized the hippocampus and not mPFC.

We could, in principle, speculate about the nature of the control process in mPFC, but do not yet have a sufficiently articulated hypothesis to justify such a discussion. We elected therefore, to not speculate about this, at this stage.

Author Action Taken 3.9. The reviewer's reasonable comments illustrated for us that we have not been sufficiently clear in articulating our assumptions about how the proposed mechanism works—e.g. that the sites of representations are likely to be elsewhere and influenced by top-down control signals from mPFC. We apologise for this lack of clarity about our proposed mechanism. We also recognize that we have more work to do to test this anatomical model, and

that we can really only speculate at this stage about how it could work. We therefore added more explicit statements about our assumptions and identified these clearly as speculation. The location and nature of these changes are detailed in Author Action Taken 3.3.

Reviewer Comment 3.10. Specific comments:

Results - the full statistical results from the ANOVA and repeated measures ANOVAs including degrees of freedom, as well as effect sizes should be presented for each finding in each experiment (i.e. line 234), rather than just presenting the p value. Also, the exact p-values should be presented rather than i.e. $p < 0.05$. See related comment for ANOVAs in Experiment 7.

Author Response 3.10. The reviewer is quite right about this. We're sorry that we did not report the requisite detail.

Author Action Taken 3.10. We have now reported full statistical results for the ANOVAs, effect sizes and exact p values for all the experiments.

Reviewer Comment 3.11. Figure: It may be a result of format changes when the files were converted for submission, but the figures were difficult to read (blurry legends), and the c-Fos staining was not visible. The resolution of the figures should be improved.

Author Action Taken 3.11. Thanks for pointing this out. We have updated the figures and hopefully we have rectified the problem. If not, please let us know and will sort out an alternative solution.

Reviewer Comment 3.12. Minor points:

The methods should be revised for repetition - for example, it is mentioned multiple times that quantification of exploratory behavior was measured manually using hand chronometers (line 725, line 736, and in the description of each condition).

Author Action Taken 3.12. Thank you for pointing this out to us. We're sorry it was repetitious. We have revised the methods accordingly.

Reviewer Comment 3.13. Typo line 705 - ramdonly

Typo line 750 - period missing after phase 2

Typos throughout referring to figures. Should be consistent. I.e. line 263-266 -- I.e. Fig. 2A, Fig 2B, f, Fig. 2 C, Fig. 2C can be all be found. Similar findings through paper.

Typos line 201 - use decimals rather than commas in t-test results.

Author Action Taken 3.13. Thanks for taking the trouble to point these out. We have corrected all of these.

Reviewer Comment 3.14. Be consistent in the terminology used. Given the complexity of the design, it is often unclear if the terms are being used interchangeable ty to refer to the same type of object, or to a different type of object entirely.

- Familiar object, practiced object, studied object
- Novel object, unpracticed object
- Encoding phase, sampling phase

- arena, context

Author Action Taken 3.14. Yes, we can see how this could sow confusion. We're grateful for the help. We have gone through and revised to adopt a more consistent terminology, and this is a great improvement. Specifically, we now generally use the term "arena context", though, to avoid repetitiousness, we sometimes simplified to "context" or "arena". We abolished "sampling phase" altogether and now simply refer to an "encoding phase". In all cases where we are referring to a novel object, we now always use Novel object and no synonyms. We eliminated "studied object" and now refer to Old objects as consistently as possible.

References

1. Miller, R.R. & Witnauer, J.E. Retrospective revaluation: The phenomenon and its theoretical implications. *Behav Processes* **123**, 15-25 (2016).
2. Anderson, M.C. Rethinking interference theory: Executive control and the mechanisms of forgetting. *Journal of Memory and Language* **49**, 415-445 (2003).
3. Blaisdell, R., London, J. & Green, J. *The call of the wild* (Dover Publications, Mineola, N.Y., 1999).
4. San-Galli, A., Marchand, A.R., Decorte, L. & Di Scala, G. Retrospective revaluation and its neural circuit in rats. *Behav Brain Res* **223**, 262-270 (2011).
5. Anderson, M.C. & Spellman, B.A. On the status of inhibitory mechanisms in cognition: memory retrieval as a model case. *Psychol Rev* **102**, 68-100 (1995).
6. Peters, G.J., David, C.N., Marcus, M.D. & Smith, D.M. The medial prefrontal cortex is critical for memory retrieval and resolving interference. *Learn Mem* **20**, 201-209 (2013).
7. Tran-Tu-Yen, D.A., Marchand, A.R., Pape, J.R., Di Scala, G. & Coutureau, E. Transient role of the rat prelimbic cortex in goal-directed behaviour. *Eur J Neurosci* **30**, 464-471 (2009).
8. Knapska, E., *et al.* Differential involvement of the central amygdala in appetitive versus aversive learning. *Learn Mem* **13**, 192-200 (2006).
9. Guzowski, J.F., Setlow, B., Wagner, E.K. & McGaugh, J.L. Experience-dependent gene expression in the rat hippocampus after spatial learning: a comparison of the immediate-early genes Arc, c-fos, and zif268. *J Neurosci* **21**, 5089-5098 (2001).
10. Cullinan, W.E., Herman, J.P., Battaglia, D.F., Akil, H. & Watson, S.J. Pattern and time course of immediate early gene expression in rat brain following acute stress. *Neuroscience* **64**, 477-505 (1995).

11. Knapska, E. & Maren, S. Reciprocal patterns of c-Fos expression in the medial prefrontal cortex and amygdala after extinction and renewal of conditioned fear. *Learn Mem* **16**, 486-493 (2009).
12. Morgan, J.I., Cohen, D.R., Hempstead, J.L. & Curran, T. Mapping patterns of c-fos expression in the central nervous system after seizure. *Science* **237**, 192-197 (1987).
13. Bertaina-Anglade, V., Tramu, G. & Destrade, C. Differential learning-stage dependent patterns of c-Fos protein expression in brain regions during the acquisition and memory consolidation of an operant task in mice. *Eur J Neurosci* **12**, 3803-3812 (2000).
14. Albasser, M.M., Poirier, G.L. & Aggleton, J.P. Qualitatively different modes of perirhinal-hippocampal engagement when rats explore novel vs. familiar objects as revealed by c-Fos imaging. *Eur J Neurosci* **31**, 134-147 (2010).
15. Zangenehpour, S. & Chaudhuri, A. Differential induction and decay curves of c-fos and zif268 revealed through dual activity maps. *Brain Res Mol Brain Res* **109**, 221-225 (2002).
16. Anderson, M.C., Bjork, R.A. & Bjork, E.L. Remembering can cause forgetting: retrieval dynamics in long-term memory. *J Exp Psychol Learn Mem Cogn* **20**, 1063-1087 (1994).
17. Kuhl, B.A., Dudukovic, N.M., Kahn, I. & Wagner, A.D. Decreased demands on cognitive control reveal the neural processing benefits of forgetting. *Nat Neurosci* **10**, 908-914 (2007).
18. Wimber, M., Alink, A., Charest, I., Kriegeskorte, N. & Anderson, M.C. Retrieval induces adaptive forgetting of competing memories via cortical pattern suppression. *Nat Neurosci* **18**, 582-589 (2015).
19. Wu, J.Q., Peters, G.J., Rittner, P., Cleland, T.A. & Smith, D.M. The hippocampus, medial prefrontal cortex, and selective memory retrieval: evidence from a rodent model of the retrieval-induced forgetting effect. *Hippocampus* **24**, 1070-1080 (2014).

Reviewers' Comments:

Reviewer #1:

Remarks to the Author:

I think the authors make good points in regards to retrospective reevaluation, and I can see that there are differences in their procedures and this phenomenon. I appreciate that they have included these references and addressed this point in the discussion.

I do still feel that the manuscript should be simplified. It is very hard going and I really did not look forward to reviewing it again as it is such an effort to get through. This was made even harder by the fact that the authors did not make clear their changes to the manuscript after review through bold or colored typeface.

I think a major way that the authors could simplify the manuscript is to reduce the number of words they use to describe their ideas and results. For example, it seems to me that the introduction makes two key points. The first is that it may be adaptive to erase or reduce storage of memories that later become less reliable or irrelevant in a competitive manner. The second is that there are XX candidate mechanisms that underlie this effect. To me, this could be summed up in a maximum of 2 short paragraphs. The same applies to the results section. Simple points are made very wordy and tiring to follow.

I think if the authors could really succinctly describe their points it would greatly improve their paper.

Reviewer #2:

Remarks to the Author:

The authors have addressed all of my concerns. I have a suggestion for minor editing to help with clarity for readers that are not familiar with the human retrieval-induced forgetting literature. The authors might consider adding to the introduction a bit more description of the basic RIF phenomenon, the methods used to test it in the laboratory and the critical controls that have been employed previously. As it stands now, the main description of the phenomenon in the introduction is a single sentence, "For example, we have found that when people try to retrieve a past event or fact, other memories that compete with and hinder retrieval are more likely to be forgotten later on." For most readers from the rodent behavioral neuroscience community, who are likely unfamiliar with RIF literature, this does not adequately describe the essential phenomenon or the extensive foundation of human RIF that the present study builds upon. However, this comment is really just a suggestion and the authors may disagree. If so, I have no objection to the text as-is.

Reviewer #3:

Remarks to the Author:

The authors have carefully and thoroughly addressed all of my concerns. The revised manuscript is a much easier read, and will make an important contribution to the field.

Replies to Reviewer Requests

Reviewer #1 (Remarks to the Author):

Reviewer Comment 1.1. I think the authors make good points in regards to retrospective reevaluation, and I can see that there are differences in their procedures and this phenomenon. I appreciate that they have included these references and addressed this point in the discussion.

Author Response 1.1. We are glad the reviewer agrees with our arguments. We were more than happy to include the additional discussion of retrospective reevaluation to address the reviewers concerns.

Reviewer Comment 1.2 I do still feel that the manuscript should be simplified. It is very hard going and I really did not look forward to reviewing it again as it is such an effort to get through. This was made even harder by the fact that the authors did not make clear their changes to the manuscript after review through bold or colored typeface.

Author Response 1.2. We regret that the lack of colored typeface caused the reviewer some difficulties in efficiently reviewing the manuscript. We have simplified the manuscript by chopping it from 7500 words (abstract and main text) to 6045 words.

Reviewer Comment 1.3. I think a major way that the authors could simplify the manuscript is to reduce the number of words they use to describe their ideas and results. For example, it seems to me that the introduction makes two key points. The first is that it may be adaptive to erase or reduce storage of memories that later become less reliable or irrelevant in a competitive manner. The second is that there are XX candidate mechanisms that underlie this effect. To me, this could be summed up in a maximum of 2 short paragraphs. The same applies to the results section. Simple points are made very wordy and tiring to follow.

I think if the authors could really succinctly describe their points it would greatly improve their paper.

Author Response 1.3. As noted above, we have now reduced the length of the manuscript from 7500 words (abstract and main text) to 6045 words, which honors the reviewers request.

Reviewer #2 (Remarks to the Author):

Reviewer Comment 2.1. The authors have addressed all of my concerns. I have a suggestion for minor editing to help with clarity for readers that are not familiar with the human retrieval-induced forgetting literature. The authors might consider adding to the introduction a bit more description of the basic RIF phenomenon, the methods used to test it in the laboratory and the critical controls that have been employed previously. As it stands now, the main description of the phenomenon in the introduction is a single sentence, "For example, we have found that when people try to retrieve a past event or fact, other memories that compete with and hinder retrieval are more likely to be forgotten later on." For most readers from the rodent behavioral neuroscience community, who are likely unfamiliar with RIF literature, this does not adequately describe the essential phenomenon or the extensive foundation of human RIF that the present study builds upon. However, this comment is really just a suggestion and the authors may disagree. If so, I have no objection to the text as-is.

Author Response 2.1. We agree with the reviewer's point. Indeed, our description of the human retrieval induced forgetting literature is very condensed, and perhaps not the easiest to understand for animal researchers. However, honouring the reviewer's request was not possible because (a) the editor requested that we reduce the overall length of the manuscript substantially (which we have done) and (b) specifically commented that the discussion of retrieval-induced forgetting in the introduction was too long, in direct opposition to the reviewer's remark. Given these facts, we reduced the length of the section on retrieval-induced forgetting in the introduction.

Reviewer #3 (Remarks to the Author):

Reviewer Comment 3.1. The authors have carefully and thoroughly addressed all of my concerns. The revised manuscript is a much easier read and will make an important contribution to the field.

Author Response 3.1. We are happy to have satisfied all of the reviewer's concerns, and we are glad to have made it easier to read.